# A signal processing and deep learning framework for methylation detection using Oxford Nanopore sequencing

Mian Umair Ahsan [1], Anagha Gouru [1,2], Joe Chan[1], Wanding Zhou [3,4] & Kai Wang [1,4] ✉

Oxford Nanopore sequencing can detect DNA methylations from ionic current signal of single molecules, offering a unique advantage over conventional methods. Additionally, adaptive sampling, a software-controlled enrichment method for targeted sequencing, allows reduced representation methylation sequencing that can be applied to CpG islands or imprinted regions. Here we present DeepMod2, a comprehensive deep-learning framework for methylation detection using ionic current signal from Nanopore sequencing. DeepMod2 implements both a bidirectional long short-term memory (BiLSTM) model and a Transformer model and can analyze POD5 and FAST5 signal files generated on R9 and R10 flowcells. Additionally, DeepMod2 can run efficiently on central processing unit (CPU) through model pruning and can infer epihaplotypes or haplotype-specific methylation calls from phased reads. We use multiple publicly available and newly generated datasets to evaluate the performance of DeepMod2 under varying scenarios. DeepMod2 has comparable performance to Guppy and Dorado, which are the current state-of-the-art methods from Oxford Nanopore Technologies that remain closed-source. Moreover, we show a high correlation (r = 0.96) between reduced representation and whole-genome Nanopore sequencing. In summary, DeepMod2 is an open-source tool that enables fast and accurate DNA methylation detection from whole-genome or adaptive sequencing data on a diverse range of flowcell types.

DNA methylation, which is the process by which methyl groups are added to specific nucleotides of DNA molecules, represents an important epigenetic change to the genomes of human and other species. Examples of DNA methylation include 5-methylcytosine (5mC), N4-methylcytosine (4mC), 5-hydroxymethylcytosine (5hmC), and N6-methyldeoxyadenosine (6mA)[1]. DNA methylations and modifications are essential to many biological processes, including genome stability, genomic imprinting, aging, repression of transposable elements, and carcinogenesis[1]. 5-methylcytosine (5mC) is the most prevalent form of DNA methylation in humans. Genome-wide epigenetic change (global DNA 5mC hypomethylation) is a hallmark of cancer[2], which is often accompanied by local hypermethylation of tumor suppressor genes within their promoter regions[3,4]. The 5mC occurs generally within CpG dinucleotides which are concentrated in large clusters called CpG islands. In addition to being diagnostic biomarkers, DNA methylation sites are now a therapeutic target for cancer

[1]Raymond G. Perelman Center for Cellular and Molecular Therapeutics, Children's Hospital of Philadelphia, Philadelphia, PA 19104, USA. [2]Department of Biology, University of Pennsylvania, Philadelphia, PA 19104, USA. [3]Center for Computational and Genomic Medicine, Children's Hospital of Philadelphia, Philadelphia, PA 19104, USA. [4]Department of Pathology and Laboratory Medicine, Perelman School of Medicine, University of Pennsylvania, Philadelphia, PA 19104, USA. ✉e-mail: wangk@chop.edu

with several drugs being tested or approved by the US Food and Drug Administration (FDA)[5,6]. For example, 5-Aza-2'-deoxycytidine is among the first methylation inhibitor used in cancer clinical trials[7], and we demonstrated that it leads to isoform switching and exon skipping such as *EZH2* in addition to de-methylation[8]. Both local 5mC hypermethylation and global 5mC hypomethylation can distinguish cancer cells from normal cells, making methylation a potential biomarker for such cells[9]. N4-methylcytosine (4 mC), 5-hydroxymethylcytosine (5hmC) and N6-methyladenine (6mA) also play pivotal role in regulating gene expression[1,10], but they are much less studied than 5mC, partially due to lack of reliable high-throughput methods and reference data sets.

Methylation microarrays and short-read sequencing have enabled profiling of 5mC in CpG sites at single base resolution, however, both technologies have substantial limitations[11]. These methods typically use bisulfite conversion[12] for 5mC detection which requires special DNA preparation[13,14], is subject to conversion efficiency[15] and PCR biases[16], has limitations to assay repetitive regions (as shown in Fig. 1) and epihaplotypes (haplotype-specific methylation), which are areas that can be addressed by long-read sequencing technologies such as Oxford Nanopore Technologies (ONT) sequencing. Ionic current

signal from Nanopore sequencing can be used to distinguish between unmethylated and methylated cytosines and several tools have been developed for this purpose (for example, Nanopolish[17], f5c[18], Megalodon[19], Dorado[20], Guppy[21], DeepMP[22], Nanoraw/Tombo[23], DeepSignal[24], Rockfish[25], as well as DeepMod[26]). Indeed, a recently published comprehensive survey evaluated seven ONT methylation callers in diverse genomic contexts and showed a high level of concordance between ONT methylation prediction and bisulfite sequencing[27].

Moreover, adaptive sampling in Nanopore sequencing allows real time DNA molecule selection based on read sequence[28]. This strategy results in a real-time acceptance or rejection of molecules: within 1 s (~400 bp bases), the sequencer determines whether to continue sequencing a molecule (if it maps to a pre-specified region of interest) or reject and then sequence the next molecule. Therefore, one can design target genomic regions such as CpG islands or CpG rich promoters, and then perform adaptive sequencing on the ONT platform, essentially achieving reduced representation methylation sequencing (RRMS) which is conceptually similar to the commonly used reduced representation bisulfite sequencing (RRBS). In 2022, Nanopore released more mature protocols of adaptive sequencing on human

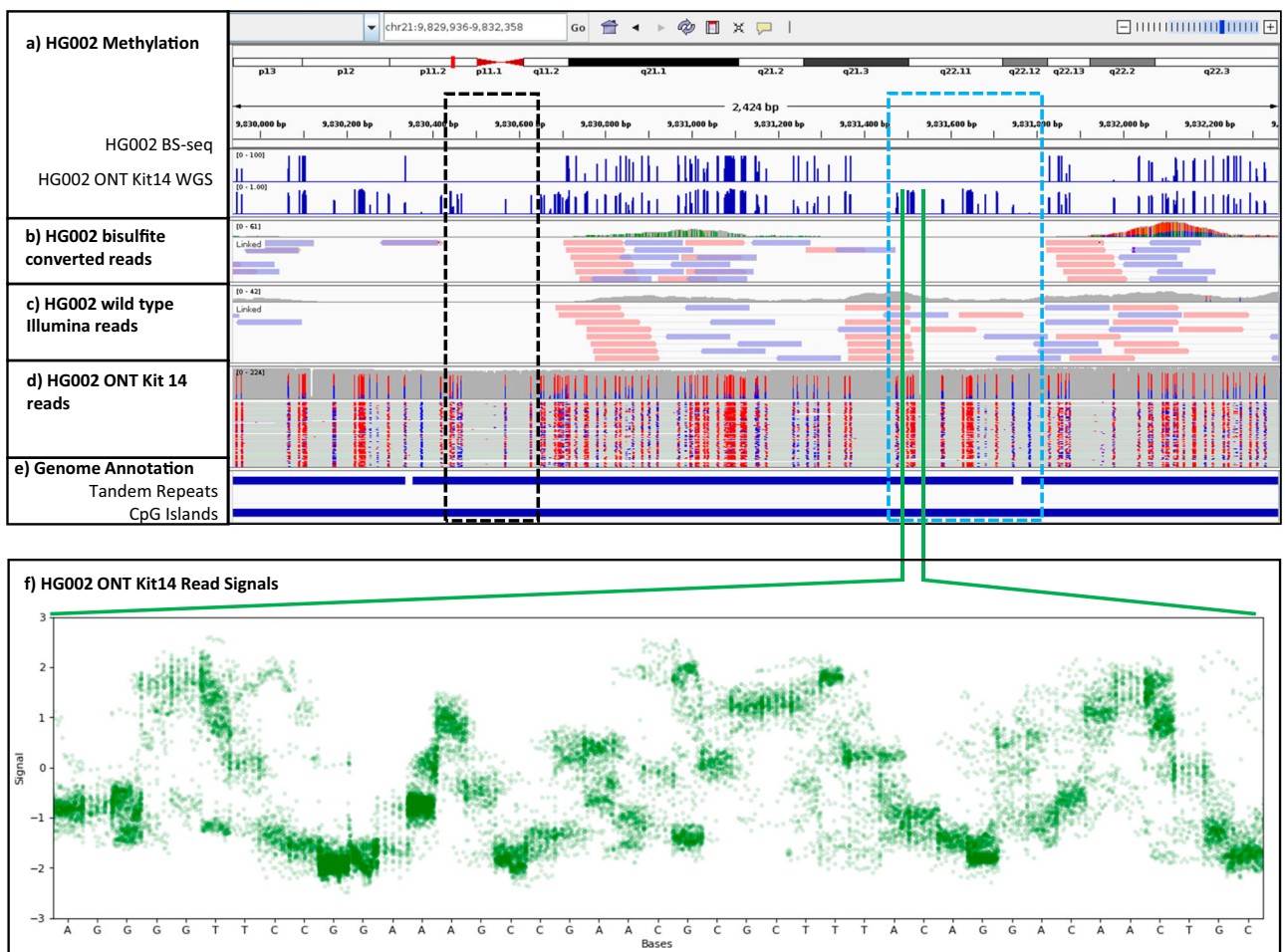

**Fig. 1 | Methylation calls and reads of short-read bisulfite sequencing (BS-seq) and Oxford Nanopore Technologies (ONT) kit14 sequencing of HG002 genome in CpG islands overlapping large tandem repeats. a** Shows BS-seq methylation calls and ONT kit14 methylation calls produced by DeepMod2, (**b**) shows IGV plot of reads and coverage of HG002 Illumina reads after bisulfite conversion, (**c**) shows IGV plot of reads and coverage of wild type HG002 Illumina reads. **d** Shows coverage and methylation tags of ONT kit14 reads of HG002 tagged by DeepMod2, with red and blue colors showing methylated and unmethylated CpGs, respectively. **e** Shows a track of tandem repeats larger than 10 kbp from GIAB genome stratification v3.0, and a track of CpG islands. The region surrounded by black rectangle shows no read coverage in either BS-seq or native Illumina sequencing due to low complexity. The region surrounded by blue rectangle has read coverage in native Illumina sequencing, but not in BS-seq sample which illustrates further difficulty of accurately mapping low complexity bisulfite converted reads. **f** Shows ONT signal of all reads overlapping the methylated CpG sites shown in the green section in (**e**). Signals from all the reads are aligned to the reference genome using move table and minimap2, with overlapping signals from all reads are shown for each base in the region. Panels (**a**–**e**) are generated in IGV.

genome (targeting 310 Mbp or 10% of the genome consisting of all CpG islands, CpG shelves, CpG shores and several promoter regions), and a few papers demonstrated its success[29–31].

Over the past few years, peer-reviewed and open-source methylation detection tools have severely lagged technological developments in ONT sequencing. For example, Nanopolish, DeepMP, DeepSignal, DeepMod, Tombo, methBERT and Rockfish provide models only for R9.4 flowcells, which is being discontinued by ONT in 2023. The newer generation of R10.4 flowcells have a longer protein pore with two pinch points for two signal measurements, and the signal profile is completely different from previous flowcells. Moreover, none of the open-source tools can process POD5 files or BAM move tables that replace FAST5 file format for storing signal and basecall data by ONT basecallers.

In the current study, we present DeepMod2, a comprehensive deep learning framework for methylation detection from Oxford Nanopore sequencing. DeepMod2 substantially improves upon our previous tool DeepMod in terms of accuracy, and it can analyze all types of Oxford Nanopore flowcells and signal data formats. We perform whole-genome Nanopore sequencing of the NIH3T3 cell line (a widely used mouse reference cell line for methylation studies) and RRMS of HG002 (a widely used human reference cell line). Using these datasets along with publicly available ONT datasets of Ashkenazim trio (HG002, HG003, HG004) human cell lines, we evaluate the performance of DeepMod2 with short-read sequencing and methylation microarrays as ground truth. Our evaluation on both R9.4 and R10.4 flowcell datasets demonstrates that DeepMod2 has comparable performance to Guppy and Dorado, the current state-of-the-art methods from Oxford Nanopore Technologies. For per-read and per-site evaluation on human cell lines, DeepMod2 achieves ~95% and ~99% F1-score, respectively, with a correlation of $r > 0.95$ with short-read sequencing. Moreover, we show a high correlation ($r = 0.96$) between reduced representation and whole-genome Nanopore sequencing of HG002, suggesting that it can be a viable strategy for large-scale cost-effective methylation profiling of complex genomic regions. Finally, we demonstrate that phased methylation calls from DeepMod2 can be used to accurately predict imprinted regions in human cell-lines.

## Results

DeepMod2 takes ionic current signal from POD5/FAST5 files and read sequences from a BAM file as input and makes 5mC methylation prediction for each read independently using a BiLSTM or Transformer model. If aligned reads are provided as input, then DeepMod2 combines per-read predictions to estimate overall methylation level for each CpG site in the reference genome. It additionally provides haplotype-specific methylation counts if the input BAM file is phased. Finally, it adds standardized methylation tags (MM and ML) to the BAM file to allow allele-specific analysis or visual validation of the methylation. These tags can be viewed in genome browsing tools, such as Integrative Genomics Viewer (IGV)[32] shown in Fig. 1. The workflow of DeepMod2 is shown in Fig. 2. Since R9.4 and R10.4 flowcells produce different characteristic signals, we present benchmark performance of DeepMod2 models trained for both types of flowcells, with one BiLSTM and one Transformer model trained for each flowcell type. We show DeepMod2's performance on reduced representation methylation sequencing (RRMS) samples of HG002 and compare it with whole genome sequencing. We also demonstrate how phased methylation output of DeepMod2 can enable detection of imprinted regions. Finally, we present a computational runtime analysis of DeepMod2 and examine the model differences between DeepMod and DeepMod2.

We compared per-read and per-site performance of DeepMod2 and other state-of-the-art Nanopore methylation callers on R9.4.1 and R10.4.1 flowcell datasets of Ashkenazim trio (HG002, HG003, HG004) and R9.4.1 flowcell dataset of NIH3T3. The evaluation was carried out against ground truth methylation labels obtained from short-read

sequencing coupled with bisulfite or enzymatic conversion. For NIH3T3, we additionally evaluated per-site performance against Infinium Mouse BeadChip methylation calls.

### Per-read benchmark evaluation

DeepMod2 and other state-of-the-art methylation callers for nanopore sequencing predict CpG methylation probability for each DNA molecule independently by analyzing the current signal it produces. Then, they combine the individual read evidence to infer methylation stoichiometry for each CpG site. It is important to directly evaluate the accuracy of the first step because the underlying models are often trained (usually through machine learning or deep learning) to maximize the accuracy of this prediction. This can be accomplished via per-read evaluation in which we compare the individual read predictions for a CpG site against the ground truth labels. In this case, the benchmark ground truth only included sites that were almost completely methylated (≥90% methylation frequency) or unmethylated (<10% methylation frequency) with ample coverage (≥10X) from short-read sequencing for this evaluation; details of this evaluation can be found in the Methods section.

We evaluated DeepMod2, Nanopolish, Guppy and Rockfish on R9.4.1 flowcell datasets of Ashkenazim trio and NIH3T3. We recognize that Megalodon and Tombo (other software tools released by ONT) are widely used in literature, however, both tools have been deprecated by ONT in favor of Dorado (performance comparison with Dorado is shown under the runtime evaluation section). For R10.4.1 flowcell data of the Ashekazim trio, we evaluated DeepMod2, Guppy and f5C (a re-implementation of Nanopolish) since these are the only tools compatible with R10.4.1 datasets. For the Ashkenazim trio, we evaluated performances on chr1 since DeepMod2 models were trained on chr2-21 and validated on chr22. Figure 3a–d show receiver operating characteristic (ROC) curves and precision-recall (PR) curves for per-read performance evaluation on HG002. The ROC and PR curves for DeepMod2, Guppy and Rockfish show that their models can achieve a good balance between methylated and unmethylated predictions as the decision threshold changes. Moreover, ROC and PR curves for R10 flowcell have much sharper corners than R9 flowcells, indicating that the signal from newer flowcells can better discriminate between methylated and unmethylated cytosines. This is corroborated by a higher area under ROC (AUROC), average precision (AP) and F1-score for R10 datasets compared to R9 datasets, as shown in Fig. 3e-j. Supplementary Data 1 shows precision, recall, F1, AUROC, AP, true and false positive and negatives of each methylation caller for each dataset. For R9.4.1 datasets of the Ashkenazim trio HG002, HG003 and HG004, the F1-score of DeepMod2 models is in the 95.7–97.1% range, which is higher than the F1-scores of Guppy models (93.17–95.31%), but lower than Rockfish (96.9–97.5%). For the mouse genome NIH3T3, although Rockfish (92.53% F1-score) and DeepMod2 Transformer (90.61% F1-score) perform substantially better than other tools (<88% F1-score), all methods perform relatively worse on NIH3T3 compared to other genomes. For R10.4.1 flowcells, DeepMod2 models have F1-scores in 97.5–98.2% range which is comparable with Guppy (F1-score 97.9–98.5%), while both tools perform substantially better than f5c (F1-score 89.2–91.4%).

### Per-site benchmark evaluation

DeepMod2 estimates per-site CpG methylation level by calculating the fraction of reads with 5mC compared to the total number of mapped bases at the locus. We performed per-site evaluation of DeepMod2 and other methylation callers using two strategies. In the first strategy, we measured precision/recall/F1-score for methylated and unmethylated CpGs genome wide and in seven specific genomic regions: CpG islands, CpG shores, CpG shelves, promoters, exons, introns, and intergenic regions. In this case, the ground truth only included sites that were mostly methylated (≥80% methylation) and mostly

# DeepMod2 Framework

**Fig. 2 | Workflow of DeepMod2 for methylation calling.** DeepMod2 uses POD5 or FAST5 files from Nanopore sequencing and a BAM file containing read sequences as input. It requires POD5/FAST5 files to be basecalled using Guppy or Dorado with move table output to allow signal alignment with the basecalls. The BAM file is recommended to be aligned to a reference genome to allow reference anchoring of signals and estimation of per-site methylation levels. DeepMod2 uses a feature extraction module to get signal summary statistics and alignment information for each base of a read in a 21-bp window centered at the CpG of interest. A feature matrix is fed to a BiLSTM or Transformer model that makes per-read predictions for each CpG site on the read. DeepMod2 adds per-read predictions to the input BAM file as modification tags and also produces a tab separated text file output containing detailed per-read predictions. In the post-processing step, all per-read predictions for a site are combined to calculate the percentage of total reads with methylated cytosine. DeepMod2 produces two outputs for per-site predictions: aggregated output in which methylation and read counts from forward and reverse strands of a CpG site are combined, and stranded output with separate counts for each strand. If a phased BAM file is given as input, then DeepMod2 additionally provides methylation counts for both haplotypes.

unmethylated (<20% methylation) with ample coverage (≥10X for HG002-4 and ≥5X for NIH3T3) from short-read sequencing for this evaluation. For Nanopore methylations callers, CpG sites with methylation frequency ≥50% were labelled as methylated and labelled unmethylated otherwise. In the second strategy, we measured Pearson's correlation coefficient between predicted methylation frequencies from Nanopore datasets and ground-truth methylation levels from short-read sequencing or beta values from methylation arrays. In this case, we included all sites from methylation array as ground truth, whereas for short-reads based ground truth, we only included sites with ample coverage (≥10X for HG002-4 and ≥5X for NIH3T3) regardless of the methylation level.

Figure 4 shows bar plots of F1-score of per-site evaluation of DeepMod2 and other methylation callers against short-reads ground truth in various genomic regions, as well as the Pearson's correlation coefficient for genome wide CpG sites. Supplementary Data 2 and Supplementary Data 3 show precision, recall, F1-score, true and false positive and negatives for R9.4.1 and R10.4.1 datasets, respectively. For R9.4.1 datasets of HG002, HG003 and HG004, DeepMod2 BiLSTM and

Rockfish consistently outperform other models, with genome-wide F1-score in a narrow range of 99.85–99.92%. All tools perform slightly better on human genomes than on the mouse genome NIH3T3, with Rockfish performing best at 99% F1-score followed by DeepMod2 at 98.75%. For R10.4.1 datasets, Guppy model shows substantial improvement over R9.4.1 model, whereas DeepMod2 models also show slight improvement. DeepMod2 BiLSTM and Guppy perform similarly across the genomes with F1-scores in the 99.94–99.97% range, whereas f5c has F1-score in 99.54–99.63% range showing comparable performance. Even though DeepMod2 Transformer and Rockfish had better per-read performance than DeepMod2 BiLSTM, this difference vanishes in per-site performance evaluation. In fact, DeepMod2 BiLSTM model consistently outperforms DeepMod2 Transformer model. Similarly, even though Nanopolish and f5c had substantially lower per-read performance (difference of 6–8%) compared to other tools, their per-site F1-score is within 0.5% of other tools.

Figure 5 shows heatmap plots and Pearson's correlation coefficient between the genome wide methylation frequency predictions by

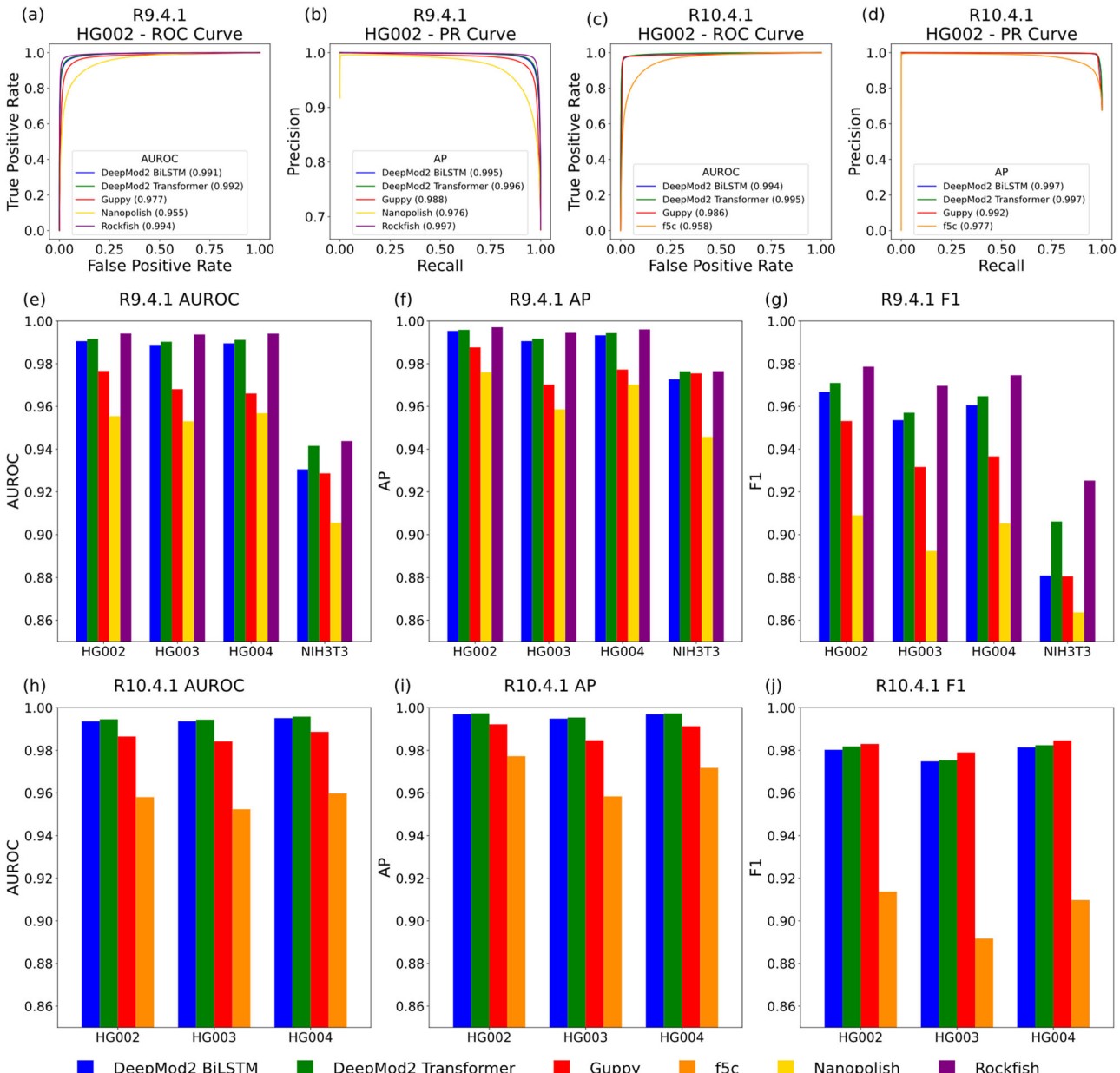

**Fig. 3 | Per-read performance of DeepMod2 BiLSTM and Transformer models and other state of the art methylation callers evaluated against short-reads ground truth for benchmarking. a–d** Show receiver operating characteristic (ROC) curves and precision-recall (PR) curves for R9.4.1 and R10.4.1 flowcell samples of HG002, with area under ROC (AUROC) and average precision (AP). **e–g** Show bar plots of AUROC, AP and F1-score for R9.4.1 flowcell datasets of HG002, HG003, HG004 and NIH3T3. **h–j** Show bar plots of AUROC, AP and F1-score for R10.4.1 flowcell datasets of HG002, HG003 and HG004. F1-score for each tool is evaluated at 0.5 probability of methylation threshold. Evaluation is performed on chr1 for HG002-4 and chr1-19 for NIH3T3. Source data are provided in the Source Data file.

Nanopore methylation callers and the ground truth methylation frequencies. For correlation and heatmap analysis, we considered only those CpG sites that were common between a given methylation caller and the ground truth after removing low coverage sites (<10X for Nanopore methylation callers and 10X/5X for Ashkenazim trio/NIH3T3 ground truth, respectively). The total number of CpG sites in the plots reflects how many CpG sites predicted by each tool had at least 10X coverage. Correlation heatmaps and marginal distributions of methylation levels show two hotspots of methylation levels in both Nanopore and short-read sequencing, corresponding to near complete methylation and non-methylation, with a smaller cluster around 50% methylation. DeepMod2 BiLSTM model shows the highest correlation (95–97.35%) across the Ashkenazim trio, whereas Rockfish has the highest correlation for NIH3T3. The heatmaps show that R10 flowcell

datasets have a better correlation with the ground truth, especially for CpG sites with intermediate methylation levels.

In our evaluation above, Nanopore methylation callers showed slightly better performance on human genomes than on mouse genome. To examine this discrepancy, we further compared Nanopore methylation calls for NIH3T3 against methylation calls from Illumina Mouse Methylation BeadChip array. Supplementary Fig. 1a shows the heatmap plots of Nanopore methylation calls against mouse methylation array. Additionally, we calculated precision, recall and F1-scores of Nanopore tools evaluated against NIH3T3 methylation microarray (329,638 CpG sites counting both strands separately), shown in Supplementary Fig. 1b, with all tools showing F1-score above 99%. These results demonstrate a high degree of consensus between methylation microarrays and Nanopore methylation calling. Furthermore, all

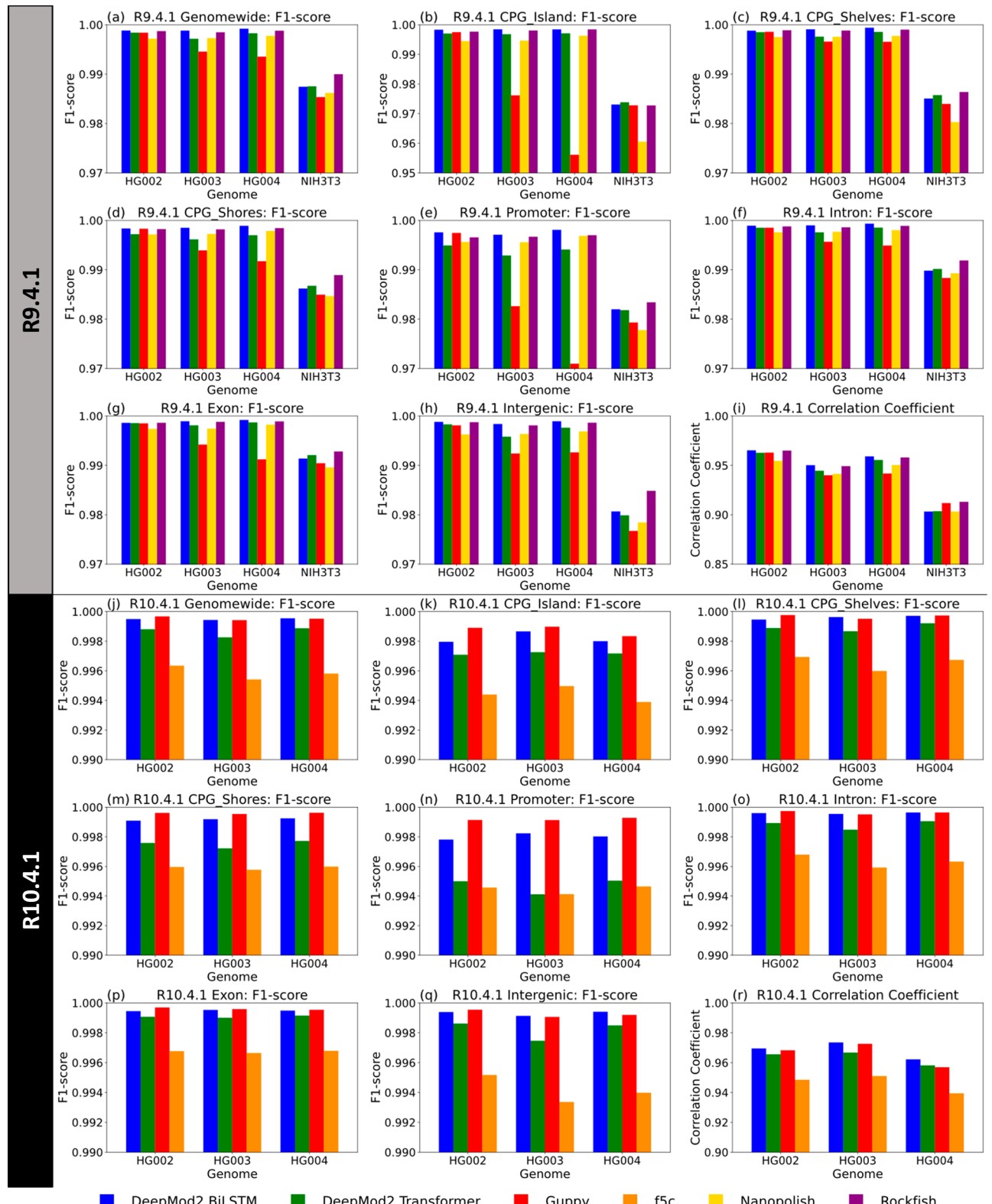

**Fig. 4 | Per-site performance evaluation of DeepMod2 and other state of the art methylation callers evaluated against ground truth. a–h, j–q** Show F1-scores of R9.4.1 and R10.4.1 datasets, respectively, from evaluation on genome wide CpG sites, as well as even key genomic regions: CpG islands, CpG shelves, CpG shores, promoters, introns, exons and intergenic regions. **i, r** Shows Pearson correlation coefficient for genome wide CpG sites for R9.4.1 and R10.4.1 datasets. Evaluation is performed on chr1 for HG002-4 and chr1-19 for NIH3T3. Source data are provided in the Source Data file.

Nanopore methylation callers show a higher correlation with methylation array than with bisulfite sequencing. This suggests that the relatively poor performance of Nanopore methods when compared against bisulfite sequencing can be partly attributed to library or sample differences between Nanopore sequencing and bisulfite-sequencing of NIH3T3.

It is important to note that all ONT methylation callers have a slight tendency to overestimate methylation levels compared to

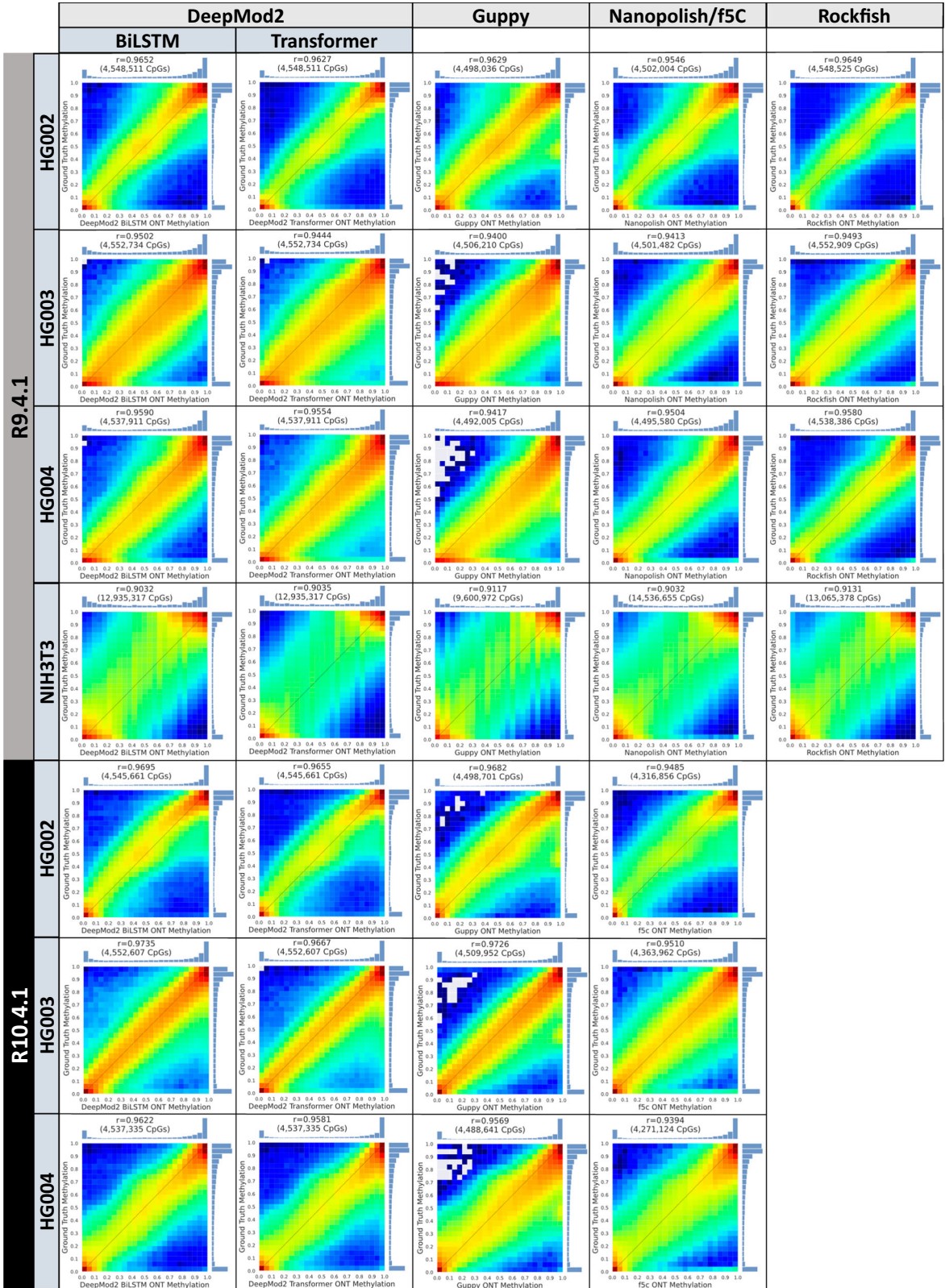

**Fig. 5 | Correlation between per-site methylation frequencies from ONT and short-read sequencing.** The figure shows Pearson's correlation coefficient (r) and correlation heatmap plot for ONT R9 and R10 flowcell datasets of DeepMod2 and other state of the art ONT methylation callers. The evaluation is performed on chr1 for HG002-4 and chr1-19 for NIH3T3. The sub-figures also show for each tool the number of CpG sites (counting forward and reverse strands separately) used to calculate correlation and plot heatmap after passing minimum coverage filter of 10X. Blue color denotes lower end of heatmap color scale whereas red represents the higher end. Source data are provided in the Source Data file.

ground truth, especially for CpG sites that have methylation frequency near 50% in ground truth. For Guppy, this tends to happen at sites where there is a "C to A/G/T" single nucleotide variant present and the overestimation can be attributed to how Modkit[33] reports Guppy methylation. Modkit reports methylation frequency of a given CpG locus as the number of reads with methylated cytosine at the locus divided by the total number of reads with cytosines at that locus. In comparison, both DeepMod2 and the ground truth report methylation frequency of a given CpG locus as the number of reads with methylated cytosine at the locus divided by the total number of reads at that locus regardless of whether the base is cytosine or not. We further investigated the cause of discrepancies between DeepMod2 and Guppy methylation frequencies in chr1 of HG002 R10.4.1 dataset. Supplementary Fig. 2a shows that once we treat SNVs as unmethylated cytosines in Guppy calls, the correlation between methylation frequencies of DeepMod2 and Guppy improves substantially. On the other hand, for 0.52% of the CpG sites, DeepMod2 predicted methylation frequency >90% while Guppy and ground truth reported methylation frequency <65%. This reveals a slight bias in DeepMod2 models to overestimate methylation in some CpG sites. Despite this, not only do DeepMod2 and Guppy have high correlation ~98% between per-site methylation frequencies, but they also have a high correlation of 93.6% between per-read probability score predictions. The heatmap and correlation of per-read probabilities is shown in Supplementary Fig. 2b which also shows the distribution of per-read probabilities from both tools. We further focused on per-read predictions of the CpG sites that were hypermethylated in DeepMod2 relative to Guppy and compared the per-read probabilities of DeepMod2 and Guppy, as shown in Supplementary Fig. 2c. For these sites, we found that the underlying per-read CpGs that had <2% probability of methylation in Guppy, DeepMod2 methylation probabilities were almost uniformly spread between 50-100%. This indicates that although DeepMod2 did not predict these per-read CpGs as unmethylated, i.e. with probability <50%, it still predicted a low confidence of methylation for these per-read CpGs. One possible reason for this discrepancy could be that DeepMod2 models are trained on native methylation found in wild type samples of HG002, HG003 and HG004 cell lines, and despite using a strict criterion for ground truth labels (discussed in "Methods" section), it is possible that residual heterogeneity among molecules exists, leading to a small fraction of mislabeled read-level methylation states used in training. This can introduce some challenges for supervised learning. One potential solution to overcome this issue can be to use synthetically modified and/or unmodified samples (that are completely methylated or unmethylated) for model training.

## Evaluation on reduced representation methylation sequencing (RRMS)

We sequenced HG002 genome using Nanopore RRMS sequencing. For the 310 Mbp on-target region, we sequenced 3.26 billion bases and achieved 12.5X coverage, resulting in 5-fold enrichment relative to the off-target region. Out of a total of 9.4 million reads sequenced, 89.4% of the reads were rejected due to being off-target (N50 609 bp), 1% of the reads were too short to be rejected or accepted and 9.4% were accepted as on-target (N50 5.41 kbp).

Figure 6a–d show IGV plots of reads and methylation calls from whole genome and RRMS Nanopore sequencing of HG002, as well as short-reads based methylation calls. Comparison of read coverage with RRMS track shows substantial enrichment for on-target regions with a sharp drop in coverage outside the target regions. We used DeepMod2 to detect 5mC from HG002 RRMS; genome-wide we detected a total of 39 out of 58.3 million cytosines within CpG motifs (i.e. counting both strands separately), whereas in the RRMS target region we detected 13.99 out of 14.37 million cytosines within CpG motifs. After aggregating the CpG predictions from forward and reverse strands, we detected 24.8 out of 29.15 million CpGs genome-

wide, and 7.17 out of 7.19 million CpGs in RRMS target region. We discovered a discrepancy between the number of aggregated and stranded CpG sites detected within RRMS regions: the number of stranded CpGs (14 million) was less than twice the number of aggregated CpGs (7.17 million). This shows that there are roughly 170k CpG loci that were covered only on the forward or reverse strand. Therefore, for lower coverage (<20X) samples, such as RRMS from MinION flowcells, we recommend using aggregated methylation counts to improve statistical power of any downstream analysis. Figure 6e shows the distribution of read depth for CpG sites in RRMS regions, with or without aggregating methylated and unmethylated counts from forward and reverses strands, as reported by DeepMod2.

Figure 6f–h show heatmap and Pearson's correlation coefficient of aggregated methylation frequencies for CpG sites within RRMS target regions when comparing ONT RRMS against ground truth from the EpiQC study[34] and ONT whole genome sequencing (R9.4.1 and R10.4.1 flowcells). We observed a high correlation (95.78–96.45%) between RRMS and WGS from both short-reads and ONT. Figure 6i shows whole genome view of coverage and methylation levels in HG002 ONT RRMS and ONT WGS datasets, with separate tracks for RMMS CpGs vs all CpGs, as well as the RRMS on-target regions track. Supplementary Data 6 shows a similar breakdown for each chromosome separately and we can visually observe the enrichment of on-target RRMS regions compared to off-target regions, as shown by the coincidence of RRMS track and coverage peaks.

## Haplotype specific methylation calling

If a phased BAM file with haplotype tag HP specifying read phase is given as input to DeepMod2, then it provides methylated and unmethylated read counts for each haplotype as additional columns in per-site output. Figure 7 shows IGV plots of phased HG002 ONT reads in imprinting control regions (ICRs) of *H19/IGF2* and *KCNQ1/KCNQ1OT1*, with methylation tags added to the BAM file by DeepMod2. In this dataset, the reads in both ICRs belong to the same haplotype block chr11: 1954146- 3277342 (GRCh38 coordinates), where SNV calling and phasing was performed by NanoCaller[35]. Figure 7 also shows methylation tracks of each haplotype extracted from DeepMod2 per-site output, in addition to the track of overall methylation frequency. The ICR of *H19/IGF2* (ICR1) is known to be methylated in paternal allele and unmethylated in maternal allele, whereas ICR of *KCNQ1/KCNQ1OT1* (ICR2) is known to be unmethylated in paternal allele and methylated in maternal allele[36]. This is consistent with phased methylation calling from DeepMod2 in Fig. 7 which shows that the parental allele in phase 1 is methylated in ICR1 and unmethylated in ICR2, whereas the parental allele in phase 2 is unmethylated in ICR1 and methylated in ICR2. Moreover, this allows us to infer that the phase 1 in the given haplotype block belongs to the paternal allele and the phase 2 belongs to the maternal allele, given that HG002 cell line is from a healthy individual without an imprinted disorder. Indeed, allele-specific methylation from ONT has been used to infer parental origin of alleles with substantial accuracy[37].

We further combined NanoCaller-Deepmod2 pipeline for phased methylation calling with DSS[38] to detect differentially methylated regions between parental haplotypes of HG002, HG003 and HG004. We used NanoCaller to phase R10.4.1 datasets of the HG002, HG003 and HG004 (without assigning maternal or paternal labels to the haplotypes), and then used DeepMod2 to detect methylation and extract aggregated methylation counts for each phase. Finally, we applied DSS on each genome separately to detect regions that were differentially methylated between the two parental haplotypes of each genome, i.e. putative imprinted regions. We detected 8909, 9873 and 10064 differentially methylated regions between parental haplotypes of HG002, HG003 and HG004, respectively. When compared against 21 known ICRs from Jima et al[36]., we found 19, 20 and 19 of the known ICRs in ONT datasets of HG002, HG003 and HG004, respectively,

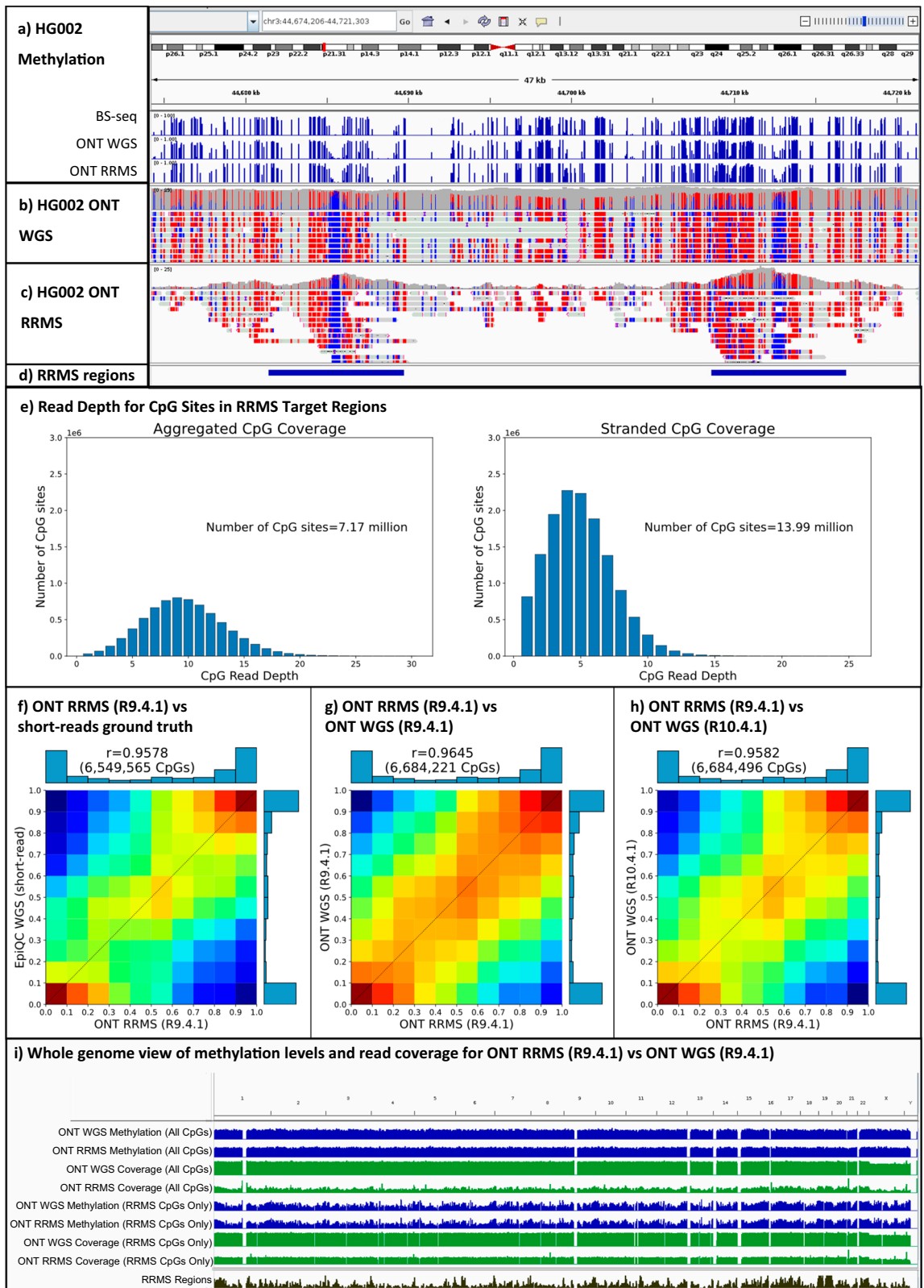

**Fig. 6 | ONT reduced representation methylation sequencing (RRMS) of HG002 genome. a–d** Show IGV plots of methylation frequencies and sequencing reads. **a** HG002 methylation levels predicted by BS-seq, ONT whole genome sequencing (WGS) and RRMS. **b** HG002 ONT WGS reads with colored methylation tags annotated by DeepMod2. **c** HG002 ONT RRMS reads with methylation tags annotated by DeepMod2. In (**b**, **c**), methylated and unmethylated cytosines are shown in red and blue, respectively. **d** Shows RRMS on-target region track. **e** Shows read depth distribution for CpG sites with or without aggregating read counts from both strands.

**f–h** Show heatmap and correlation between methylation frequencies for aggregated CpG sites from HG002 ONT RRMS and HG002 WGS from short-reads, ONT R9.4.1 and R10.4.1 flowcells, within RRMS regions of all 24 chromosomes. **f–h** Show heatmap and correlation between ONT RRMS methylation and short-read ground truth, ONT WGS R9.4.1 and ONT WGS R10.4.1 methylation of HG002. **i** Shows RRMS region track. Source data are provided as a Source Data file. Panels (**a–d**) and (**i**) are generated in IGV.

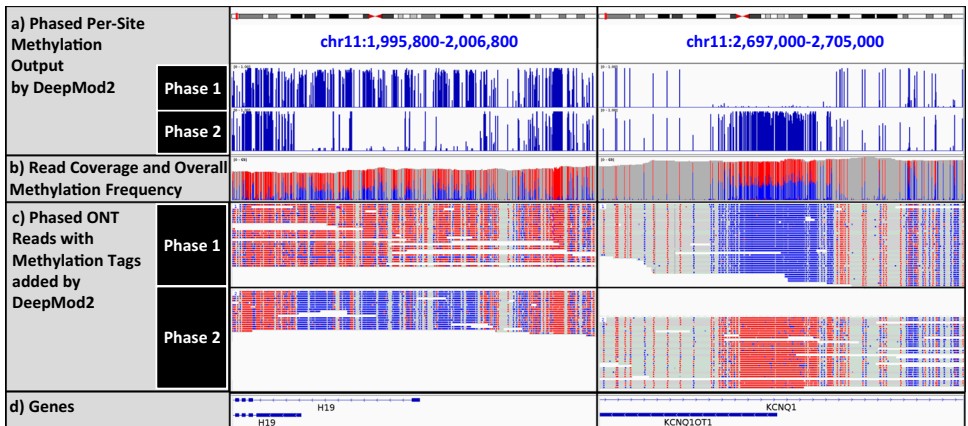

**Fig. 7 | IGV plot of phased HG002 ONT reads and methylation calls by Deep-Mod2 in two imprinting control regions (ICRs) shown in GRCh38 coordinates.** The region on the left shows the ICR of H19/IGF2 gene cluster that is methylated only in the paternal allele, whereas the region on right shows the ICR of KCNQ1/ KCNQ1OT1 gene cluster which is methylated only in the maternal allele. Reads in both regions belong to the same haplotype block with phasing done by NanoCaller. **a** Shows phased per-site output by DeepMod2, i.e. methylation frequency found in each haplotype. **b** Shows read coverage track with red and blue colors depicting the ratio of methylated and unmethylated reads, respectively, at each CpG site regardless of the phase. The methylation fraction calculated from all reads hovers in 0.4–0.6 range, but this fraction can be affected by uneven coverage between alleles. Phased methylation frequencies on the other hand Show a much more drastic difference between methylation frequencies of the alleles. **c** Shows phased ONT reads with methylation tags added by DeepMod2, with red and blue depicting methylated and unmethylated cytosines. **d** Shows RefSeq gene tracks for genes overlapping the two ICRs. Panels (**a**–**d**) are generated in IGV.

using NanoCaller-DeepMod2-DSS pipeline, as shown in Table 1. We further investigated the cause behind false negative ICRs. For ICR in chr20:37522341-37522993, none of the ONT datasets show differential methylation between haplotypes and instead show near complete methylation, as shown in Supplementary Fig. 3. Approximately 1.5 kbp upstream of this ICR, both HG002 and HG003 show differential methylation for ICR in chr20:37520202-37521842, but HG004 shows near complete methylation in both haplotypes as shown in Supplementary Fig. 4. Lastly, for ICR in chr20:43513725-43515256 near *L3MBTL1*, we detected differential methylation between haplotypes of both HG003 and HG004, but not for HG002. Upon closer examination, we found that HG002 is indeed partially methylated in this region and the average overall methylation (ignoring read phases) reported by DeepMod2 for this region is ~40%. However, this region falls within a long region (~700 kbp) of runs of homozygosity in HG002, and as a result no read phasing was performed and no phased methylation counts were produced for this region, as shown in Supplementary Fig. 5. This illustrates a drawback in depending solely upon phased methylation counts for analyzing imprinted regions, because read phasing may not be possible in regions with runs of homozygosity. Therefore, there is a need to develop differential methylation analysis methods that can incorporate both haplotype-specific methylation levels as well as overall methylation levels when analyzing differential methylation patterns in a parent-of-origin specific manner.

We also compared candidate ICRs from DeepMod2 with 1488 candidate ICRs identified by Jima et al[36]. and found 429, 346 and 377 of these candidate ICRs in ONT datasets of HG002, HG003 and HG004, respectively. Supplementary Data 7 shows all the candidate ICRs and differentially methylated regions found within haplotypes of HG002, HG003 and HG004, along with DSS area statistic for each region.

### Evaluation of DeepMod2 runtime and accuracy under various model parameters, basecaller and alignment options

DeepMod2 uses BiLSTM and Transformer-based deep neural networks for methylation detection and we recommend using a GPU for the fastest runtime. However, GPUs or GPU instances on cloud servers can be an expensive and scarce resource, whereas CPUs are often much cheaper and are more readily available. In this section we evaluated runtime performance of DeepMod2 on a single PromethION R10.4.1 flowcell dataset of HG004 with ~30X coverage. The dataset consists of 4.9 million reads in POD5 file format that were basecalled with Dorado and then aligned with minimap2. All software were run using 16 CPUs (Intel Xeon Gold 5317 3-GHz) with 256GB of memory, with NVIDIA A100 80GB GPU for running for Dorado and DeepMod2 deep-learning models. We also present the runtime for DeepMod2 using CPUs instead of GPUs for model inference.

To allow fast inference from CPUs, we applied model pruning to BiLSTM and Transformer models in DeepMod2 after training. Pruning introduces sparsity in the models by setting certain model weights to zero, essentially removing some connections between layers. While doing so has no effect on GPU runtime, it can allow tremendous speed up for CPU inference. We applied pruning to linear layers in BiLSTM and Transformer models by removing a certain percentage of weights with lowest L1-norm; more details are provided in the Methods section. This reduces the size of the BiLSTM model from 1,236,225 parameters to 582,503 parameters and reduces the size of the Transformer model from 545,121 to 374,049 parameters. However, there is a possibility that model pruning can reduce the model accuracy, and we wanted to assess its effects. Figure 8a shows the effect of model pruning on BiLSTM model runtime and accuracy. For CPU inference, the pruned BiLSTM model requires only 13 h compared to 60 h for the model without pruning, demonstrating approximately 5-fold speed up with pruning. For GPU inference, the pruned model requires only 5.7 h. The per-read F1-scores of BiLSTM model with and without pruning are 95.18% and 95.39%, respectively. Similarly, the per-site F1-scores of BiLSTM model with and without pruning are 99.86% and 99.88%, respectively. These results demonstrate that model pruning has negligible effect on model accuracy while allowing a substantial improvement in runtime. Pruning is enabled by default in DeepMod2, but users can choose to disable pruning. Supplementary Data 4 shows a detailed performance comparison of DeepMod2 BiLSTM and Transformer models with and without pruning, with evaluation shown on the entire R9 and R10 flowcell datasets.

ONT basecallers such as Guppy and Dorado provide several basecaller models for each flowcell: FAST, HAC (high accuracy) and SUP (super high accuracy), in the increasing order of basecall accuracy, model size and runtime. Although we performed per-read and per-site evaluation in earlier section on SUP basecalled datasets, we further investigated the effect of basecaller model choice on DeepMod2

**Table 1 | Detection of 21 known imprinting control regions (ICRs) from ONT datasets of HG002, HG003 and HG004 using phased methylation calls from DeepMod2**

| Known ICRs | | Imprinted Regions from ONT Phased Methylation | | | | | |
|---|---|---|---|---|---|---|---|
| | | HG002 | | HG003 | | HG004 | |
| ICR Coordinates | Nearest Gene | Coordinates | Diff | Coordinates | Diff | Coordinates | Diff |
| chr1:68049858-68051097 | DIRAS3 | chr1:68049854-68052007 | 0.82 | chr1:68049525-68052007 | 0.63 | chr1:68050054-68052007 | 0.71 |
| chr6:3848794-3850307 | FAM50B | chr6:3848793-3850420 | 0.71 | chr6:3848793-3850222 | 0.67 | chr6:3848793-3850185 | 0.66 |
| chr6:144006941-144008825 | PLAGL1\|HYMAI | chr6:144006832-144008884 | 0.87 | chr6:144006832-144008932 | 0.90 | chr6:144006707-144008932 | 0.89 |
| chr7:50781638-50783354 | GRB10 | chr7:50781904-50783353 | 0.90 | chr7:50781904-50783353 | 0.78 | chr7:50781255-50783554 | 0.86 |
| chr7:130490640-130494200 | MEST\|MESTIT1 | chr7:130490478-130493448 | 0.85 | chr7:130490104-130493840 | 0.73 | chr7:130490535-130493840 | 0.91 |
| chr11:1997886-1999417 | MRPL23\|H19 | chr11:1997669-2003426 | 0.79 | chr11:1997610-2001449 | 0.67 | chr11:1997531-2003611 | 0.82 |
| chr11:1999793-2000383 | MRPL23\|H19 | chr11:1997669-2003426 | 0.79 | chr11:1997610-2001449 | 0.67 | chr11:1997531-2003611 | 0.82 |
| chr11:2000487-2001247 | MRPL23\|H19 | chr11:1997669-2003426 | 0.79 | chr11:1997610-2001449 | 0.67 | chr11:1997531-2003611 | 0.82 |
| chr11:2001655-2003118 | MRPL23 | chr11:1997669-2003426 | 0.79 | chr11:2001890-2003556 | 0.77 | chr11:1997531-2003611 | 0.82 |
| chr11:2698157-2699485 | KCNQ1\|KCNQ1OT1 | chr11:2698550-2701209 | 0.91 | chr11:2698550-2701308 | 0.87 | chr11:2698156-2701394 | 0.90 |
| chr11:2699814-2701210 | KCNQ1\|KCNQ1OT1 | chr11:2698550-2701209 | 0.91 | chr11:2698550-2701308 | 0.87 | chr11:2698156-2701394 | 0.90 |
| chr13:48317894-48321417 | RB1\|PPP1R26P1 | chr13:48318011-48321833 | 0.90 | chr13:48318026-48321652 | 0.64 | chr13:48318337-48321625 | 0.54 |
| chr14:100824556-100828242 | MEG3 | chr14:100823703-100827916 | 0.74 | chr14:100823966-100827691 | 0.69 | chr14:100823902-100828027 | 0.70 |
| chr15:23647239-23648622 | MAGEL2 | chr15:23647058-23648621 | 0.65 | chr15:23647088-23648621 | 0.51 | chr15:23647208-23648774 | 0.66 |
| chr15:23686523-23686574 | NDN | chr15:23686339-23687716 | 0.69 | chr15:23686273-23687478 | 0.35 | chr15:23686522-23687854 | 0.65 |
| chr19:56837320-56841439 | ZIM2\|PEG3\|MIMT1 | chr19:56837124-56841579 | 0.79 | chr19:56837319-56838117, chr19:56838259-56841977 | 0.43, 0.49 | chr19:56837566-56841579 | 0.66 |
| chr20:37520202-37521842 | BLCAP\|NNAT | chr20:37520296-37522257 | 0.76 | chr20:37520201-37522340 | 0.42 | - | - |
| chr20:37522341-37522993 | BLCAP\|NNAT | - | - | - | - | - | - |
| chr20:43513725-43515256 | L3MBTL1 | - | - | chr20:43513364-43515956 | 0.78 | chr20:43513364-43515956 | 0.86 |
| chr20:58839107-58842875 | GNAS | chr20:58838772-58845170 | 0.88 | chr20:58838772-58845034 | 0.78 | chr20:58838983-58843776 | 0.84 |
| chr20:58850158-58852357 | GNAS | chr20:58850396-58852480 | 0.65 | chr20:58850396-58852480 | 0.76 | chr20:58850396-58853143 | 0.86 |

For each known ICR, its GRCh38 coordinates and the genes nearest to it are listed. For imprinted regions detected in HG002, HG003 and HG004 using ONT, the GRCh38 coordinates and the difference in average methylation levels between haplotypes (Diff) detected by DSS is shown.

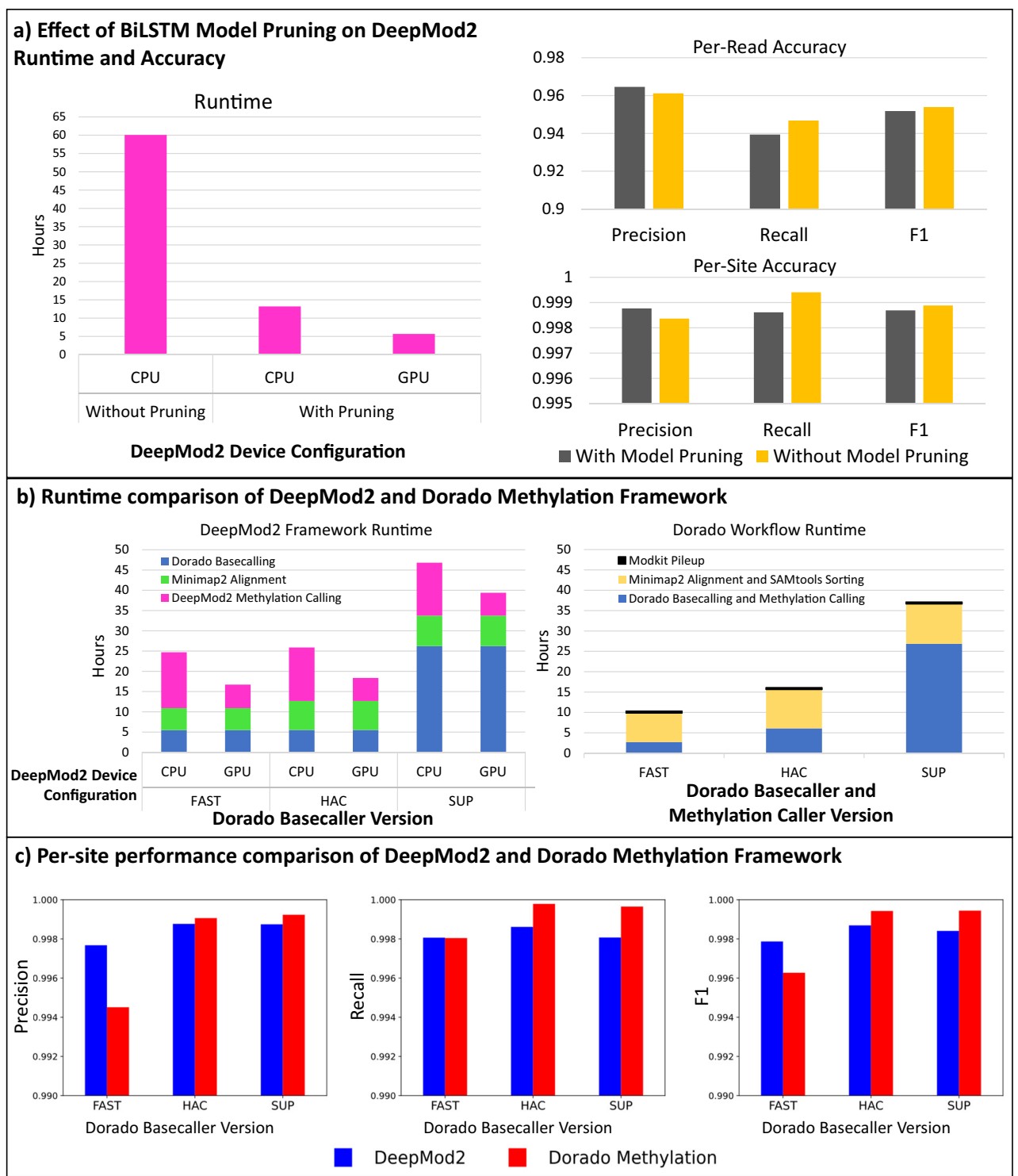

**Fig. 8 | Runtime performance of DeepMod2 and Dorado on a single PromethION flowcell dataset of HG004 using 16 Intel Xeon Gold 5317 3-GHz CPUs and 1 NVIDIA A100 GPU. a** Shows the substantial effect of pruning BiLSTM model on DeepMod2 runtime when using CPUs and GPU for model inference, as well as the negligible effect of pruning per-read and per-site accuracy. **b** Shows runtime comparison of DeepMod2 and Dorado methylation calling frameworks under three basecaller models, FAST, HAC and SUP. For DeepMod2, runtimes for CPU and GPU inference are shown separately, whereas for Dorado only GPU inference runtime is shown. **c** Shows precision, recall and F1-score of DeepMod2 and Dorado with the three basecaller models. Source data are provided in the Source Data file.

runtime and accuracy. We analyzed the runtime of the entire Deep-Mod2 methylation calling framework, which has three basic steps: (1) basecalling, (2) read alignment, and (3) methylation calling, and compared it with Dorado methylation calling framework which has three similar steps: (1) basecalling and methylation calling, (2) read alignment, BAM sorting and indexing, and (3) per-site pileup from BAM

file. Since Dorado integrates basecalling and methylation into a single run, we used GPU for running all instances of Dorado, with or without methylation calling. For DeepMod2, we compared runtime between two device configurations for model inference: using CPUs vs GPU for model inference, and the results are shown in Fig. 8b and Supplementary Table 1. In DeepMod2 framework, Dorado basecalling using

both FAST and HAC basecaller models takes ~5.5 h, whereas SUP model takes 26 h. Afterwards, using minimap2[39] for genome alignment takes 5.4 h for FAST basecalled reads and 7.2–7.5 h for HAC and SUP basecalled reads. For DeepMod2 pruned BiLSTM model, regardless of the basecaller model, CPU inference takes ~13 h and GPU inference takes only ~5.6 h. Therefore, for the end-to-end pipeline from raw signals to per-read and per-site predictions and methylated tagged BAM, DeepMod2 framework with CPU inference takes 25, 26 and 47 h for FAST, HAC and SUP basecalling, respectively, whereas GPU inference takes only 17, 18, and 39 h for FAST, HAC and SUP basecalling, respectively. Dorado framework on the other hand with GPU inference for both basecalling and methylation calling, takes 10, 16 and 37 h for FAST, HAC and SUP basecalling, respectively.

In terms of genome-wide per-site accuracy shown in Fig. 8c, using HAC vs SUP basecalling has little effect on DeepMod2 and Dorado methylation calling performance: F1-score for HAC basecalling is 99.86% for DeepMod2 and 99.94% for Dorado, and whereas for SUP basecalling the F1-score is 99.84% for DeepMod2 and 99.94% for Dorado. On the other hand, for FAST basecalling, we see a small drop in performance with F1-scores of 99.78% for DeepMod2 and 99.62% for Dorado. Supplementary Data 5 shows detailed per-site evaluation performance of DeepMod2 and Dorado with different basecaller models in various genomic regions.

### Reference free methylation detection

By default, when an aligned BAM is provided as input to DeepMod2, both mapped and unmapped reads are analyzed for methylation calling. Even for mapped reads, any clipped or unaligned segments are also analyzed and tagged with methylation in BAM output. This can be very useful in cases where the reference genome for a given species is not available or when a query sample differs substantially from the available reference genome due to structural rearrangements (such as chromosomal abnormalities in cancer). Supplementary Fig. 6 shows a 5281-bp insertion in HG002 genome, and all ONT methylation callers, except for DeepMod2 and Guppy/Dorado, ignored the inserted sequence and called methylation only in the read segments that aligned to the reference genome. However, DeepMod2 and Guppy/Dorado tag all CpGs in a read, regardless of the alignment, and save the methylation calls in the primary alignment record of the read in BAM output. Both tools do not add methylation tags to supplementary or secondary alignments, which can be unreliable unless strict filters are applied. Later, the BAM file can be re-aligned to different reference genomes or contigs to assess the methylation without the need to re-run methylation inference. Supplementary Fig. 6 also shows that after re-aligning methylation tagged DeepMod2 reads to the 5218-bp insertion sequence, we can examine methylation inside the insertion. Similarly, Supplementary Fig. 7 shows an example of a 1200 bp heterozygous inversion in HG002 where realignment to the inverted sequence allows us to examine methylation within the inversion.

To ensure that methylation calling without reference alignment can still provide accurate results, we investigated the accuracy of DeepMod2 when an unaligned BAM file is provided as an input. This means that DeepMod2 will only call methylation on CG motifs found on the reads and will not use reference sequence as a feature in the deep-learning model. After running DeepMod2, we mapped the unaligned methylation tagged BAM output of DeepMod2 to reference genome with minimap2 (while preserving the methylation tags MM and ML) and used Modkit[33] to get per-site frequencies. For single R10.4.1 PromethION flowcell dataset of HG004, the F1-score for methylation calling from unaligned BAM file is between 99.5–99.8% for the three basecaller versions and is within 0.3% of F1-score of DeepMod2 methylation calling with aligned BAM and reference features. Supplementary Data 5 shows a detailed performance breakdown. This shows that even without using reference anchored methylation calling or reference sequence features, DeepMod2 still achieves very high

accuracy. Thus, DeepMod2 can reliably call methylation, even when no read alignment or reference is available.

### Comparison of DeepMod2 with DeepMod

Our group previously published DeepMod for 5mC detection from Oxford Nanopore sequencing. DeepMod2 uses different neural network architectures than DeepMod and employs a different algorithm for signal processing and feature extraction. DeepMod's deep-learning model uses three BiLSTM layers of hidden size 100, after which it applies a single fully connected layer (with 400 weights) only to the central timestep (corresponding to the cytosine of interest) to obtain the probability of modification. In comparison, DeepMod2's BiLSTM model uses two BiLSTM layers of size 128, after which it applies a fully connected layer (with ~700k weights) to the concatenated output of all BiLSTM timesteps, not just the central timestep. Then it applies another fully connected layer (with 129 weights) to obtain the probability of modification. In short, DeepMod2 uses shallower recurrent network than DeepMod but uses a deeper classification network that incorporates information from all time steps. The addition of a fully connected layer of ~700k which connects to all timesteps allows for much faster convergence of model compared to using on the central timestep. However, to enable fast inference from DeepMod2 BiLSTM models despite the increased size, we applied pruning to the ~700k weight layer and removed 95% of lowest L1-norm weights. Analysis of the weights after pruning shows that the model retains at least 4% of weights from all timesteps of the concatenated BiLSTM outputs, not just the central timestep, as shown in Supplementary Fig. 8. This lends weight to the hypothesis that DeepMod2 classifier network uses some information from all timesteps in the final prediction, not just the central timestep. After pruning, DeepMod2 model has 582,503 parameters compared to 408,402 parameters in DeepMod. Moreover, DeepMod2 also implements a Transformer model which replaces the BiLSTM layers with 4 Transformer encoder layers with 8 self-attention heads.

We ran DeepMod on chr21 of HG001 R9.4 dataset basecalled with Metrichore from Nanopore WGS consortium[40]. We also basecalled the same dataset with Guppy and used DeepMod2 R9.4.1 BiLSTM model. Per-site evaluation of DeepMod had an F1-score of 87.6% and a correlation of 72.4% whereas DeepMod2 had an F1-score of 99.1% and a correlation of 91.3%. To improve methylation calling performance, DeepMod applies an optional clustering BiLSTM network (10,623 parameters), which uses predicted per-site methylation frequencies of the opposite strand and the nearby CpG sites to re-estimate the true underlying methylation frequency of a given CpG locus. This essentially has the effect of nudging the methylation frequency of a CpG site in the same direction as the nearby sites. After applying this secondary network, the average methylation for chr21 by DeepMod increased by 16.5%, the F1-score increased to 97.8% and the correlation increases to 85.6%; this is still lower than DeepMod2 which does not use any such clustering network. Moreover, the methylation frequency predictions of the second clustering network of DeepMod do not reflect true molecular stoichiometry. For instance, if DeepMod's first BiLSTM network predicts 7 out of 10 reads to be methylated at a CpG site, giving 70% methylation frequency, then after applying the secondary network, it could change the methylation frequency to 83%. It is not clear how to interpret an 83% methylation frequency out of 10 reads as it does not represent the actual stoichiometry of DNA fragments, nor does it tell which of the per-read predictions should be updated to reflect the newly estimated methylation frequency. This is certainly a problem for analyzing real-world samples drawn from patients or tumors that can have high mosaicism or heterogeneity where only a portion of cells may have aberrant methylation. Similarly, this approach is not suitable if there is a need to analyze allele-specific methylation where it is important to know the methylation status of reads containing a certain allele, not just the overall methylation

frequency of a locus. This highlights the importance of increasing the accuracy of deep-learning model at individual read level instead of relying on methylation frequency consensus from nearby sites. To that end, our results demonstrate that DeepMod2 models have substantially higher accuracy than DeepMod, due to improved model architecture and a larger training dataset.

With regards to feature extraction, both DeepMod and DeepMod2 extract features within a 21-bp window centered at the cytosine of interest. However, DeepMod2 uses 19 features to better capture the characteristics of the signals associated with each base. For instance, DeepMod2 divides the signal of each base into quadrants and calculates the mean of each quadrant to be used as a feature; it also calculates median of the signal which is more robust to poor signal alignment. DeepMod aligns the Nanopore signal against reference genome using read alignment and ignores read sequence afterwards, whereas DeepMod2 models use both read and reference sequence as features. More importantly, DeepMod2 models are trained to detect modification even if there is no read alignment to the reference, allowing it to detect modifications from unaligned reads or unaligned segments of a reads or insertions. Supplementary Table 2 goes into further details regarding the differences between the algorithm and models of DeepMod2 and DeepMod, whereas Supplementary Table 3 shows detailed breakdown of per-site performance of DeepMod and DeepMod2 on chr21 of HG001.

## Discussion

### Summary of benchmark evaluation

DeepMod2 allows fast and accurate detection of 5mC modification from ionic current signal of Oxford Nanopore sequencing. It can process POD5 and FAST5 files with signal alignment information produced by Guppy or Dorado basecallers, and it provides models for analyzing both R10.4 and R9.4 series flowcells. DeepMod2 produces detailed per-read and per-site predictions as well as methylation tagged BAM file as outputs. Our evaluation of DeepMod2 using ground truth from short-read sequencing and methylation microarray shows that Deep-Mod2 performs competitively against other state of the art methylation detection tools such as Guppy, Dorado and Rockfish, with all tools often performing within 0.2% of each other in terms of per-site F1-scores. Our analysis of both R10.4 and R9.4 datasets show a substantial improvement in methylation calling accuracy using the newer R10.4 flowcells. We performed whole genome Nanopore sequencing of mouse genome NIH3T3 and our evaluation demonstrates that Deep-Mod2 models trained on human genome can be successfully applied to the genomes of other species. Lastly, we performed differential methylation detection on haplotype-specific methylation calls from DeepMod2 to detect putative imprinted regions in HG002, HG003 and HG004 genomes. Our results showed a substantial overlap with previously known imprinting control regions (ICRs), detecting 19 out of 21 known ICRs in all three genomes.

### Application of adaptive sampling to methylation calling

Besides whole genome sequencing, reduced representation methylation sequencing (RRMS) via adaptive sampling in Oxford Nanopore sequencing allows an easy way to target and enrich certain genomic regions for methylation detection while avoiding several problems faced by RRBS, such as low coverage in regions lacking CCGG motif targeted by endonuclease. We performed reduced representation methylation sequencing (RRMS) of HG002 genome and using Deep-Mod2 we demonstrated a high correlation (>95%) between RRMS and ONT/short-read whole genome sequencing. Using a single MinION flowcell, we were able to detect 7 million CpG sites with ~12.5X coverage in 310 Mbp on-target region, covering about 10% of the human genome. This allows a substantially broader examination of DNA methylation than microarray-based methods such as Illumina 450 K, Epic V1 and EPIC V2 methylation arrays which target approximately

450K, 850K and 930K CpG sites respectively. Furthermore, unlike microarrays, adaptive sampling is an easily customizable process which allows users to target different genomic regions by simply providing a different list of acceptable genomic sequences or regions to Oxford Nanopore sequencer. This can allow development of different adaptive sampling procedures tailored towards different diseases. For example, to analyze imprinted disorders, target regions consisting of imprinted genes and imprinting control regions can be used, whereas different oncogenes and tumor suppressor genes can be targeted for different types of cancers.

### Notable features and advantages of DeepMod2

DeepMod2 shares several advantages with Guppy and Dorado over other open-source ONT methylation callers such as Nanopolish, Rockfish, f5c, DeepSignal and methBERT. For instance, DeepMod2 can store methylation information for each CG motif on a read in MM and ML tags of its BAM output. Sorted and indexed BAM files allow fast querying into arbitrary genomic positions, enabling convenient visual validation of methylation (using genome browsers such as IGV) and allele-specific analysis of methylation. Whereas other open-source tools only produce a plain-text per-read output that typically contains several hundred million lines of unordered predictions, making it difficult to assess methylation of different reads from the same region. More importantly, both DeepMod2 and Guppy/Dorado can detect methylation from unmapped reads or unaligned segments of reads and store this methylation information in the BAM file. In fact, both DeepMod2 and Guppy/Dorado do not require reference genome or aligned reads to accurately call 5mC methylation, as shown earlier in the runtime analysis. Methylation-tagged BAM files can be easily aligned to various reference genomes without the need to redo methylation detection with respect to each reference genome. This is possible because methylation tags annotate 5mC with respect to read coordinates and thus remain unchanged during alignment. This can be especially helpful in cases where the reference genome for a certain species has large gaps or is inaccurate, or the only reference sequence available is from a closely related species. Moreover, the presence of structural variants and chromosomal rearrangements frequently lead to incomplete or inaccurate read alignments, and certain methylation patterns may only become visible after aligning to a modified or different reference genome, as shown in Supplementary Fig. 6 and Supplementary Fig. 7. In particular, the emergence of pangenome references highlights the need to extricate methylation detection from reference alignment.

However, Nanopolish, f5c, Rockfish, DeepSignal, methBERT and the original DeepMod can only predict methylation for read bases that align to a reference CpG locus, and none of these tools produce a BAM file output. As a result, these tools are unable to carry out de novo methylation detection without a reference genome, and need to perform methylation from scratch if the reference genome is changed. DeepSignal and methBERT depend upon Tombo (a deprecated tool by ONT that works only for R9.4.1 flowcells) to explicitly align the read signal against a simulated reference signal using dynamic time warping algorithm. Consequently, they are unable to provide coordinates of methylated cytosines on the read since Tombo only reports the segments of raw signal that correspond to each reference base. Rockfish, on the other hand, internally uses minimap2's python API to map basecalled sequences from FAST5 files to a reference genome and uses the mapped reference sequence as input for its decoder. On the other hand, DeepMod2 and Guppy/Dorado methylation calls can be realigned to various reference genomes or split across alleles and haplotypes using well known BAM utility tools such as Modkit. However, it should be noted that de novo detection of 5mC methylation is largely possible because the ONT basecallers are able to accurately detect the underlying cytosine base regardless of 5mC methylation status. This requires incorporation of a diverse range of modified and unmodified

datasets during basecaller training. For other types of modifications, especially any rare modifications that the basecaller has not been trained on, accurate modification detection may still require reference mapping to overcome basecalling errors.

Lastly, DeepMod2 provides a convenient framework for haplotype specific methylation calling. If given a phased BAM file as input, DeepMod2 can provide methylation counts in each haplotype without any incurring extra runtime or complicated FAST5/POD5 or BAM operations. Even if DeepMod2 is given unphased BAM file as input, the methylation-tagged BAM file it produces can still be phased later. Afterwards, the phased BAM file can be split into separate BAM files for each phase, and haplotype-specific methylation calls can be extracted by running tools such as Modkit (same can be done for Guppy and Dorado methylation-tagged BAM files). Performing haplotype-specific methylation calls from other open-source methylation tools such as Rockfish, DeepMod, DeepSignal, and methBERT would require a cumbersome process of (1) splitting FAST5/POD5 files that contain raw signal data for thousands of reads per file, (2) saving the raw signal data of reads from each phase into separate FAST5 files; both steps can end up using several terabytes of storage and hours of runtime. In the case of Nanopolish and f5c, haplotype-specific methylation calling would require providing haplotype-separated BAM files as inputs in different runs. Either case ends up incurring substantial storage and runtime.

DeepMod2 also has a few advantages over ONT proprietary software such as Guppy and Dorado. Firstly, Guppy and Dorado combine basecalling and methylation calling into a single step which can save substantial time by eliminating the need to reopen the signal files and process the signals for methylation (as required by DeepMod2). However, if one needs to re-analyze the same sample for a different type of methylation (6mA/5hmC/4mC) or synthetic DNA modification (IdU/BrdU), or analyze DNA methylation in a different motif, then Dorado/Guppy needs to re-basecall the sample from scratch. This can lead to substantial redundancies in the usage of computational resources as basecalling is a very resource intensive task which typically requires high-end GPUs[41], whereas methylation or modification calling is a relatively simpler task which can be efficiently and quickly performed even on CPUs as shown by DeepMod2. Currently in Dorado v0.3.4, there are four distinct methylation detection models provided for "dna_r10.4.1_e8.2_400bps_sup@v4.2.0" basecaller model: "5mCG_5hmCG" for 5mC and 5hmC detection in CpG motif, "5mC_5hmC" for 5mC and 5hmC detection in all context, "5mC" for 5mC detection in all contexts, and "6mA" for 6mA detection in all contexts. All of these four methylation models are distinct and not necessarily interchangeable, e.g. 5mC model uses 2-class classification between 5mC and C, whereas 5mC_5hmC models uses 3-class classification between 5mC, 5hmC and C. Running more than one of these Dorado methylation models, e.g. 5mCG_5hmCG and 6mA, would require basecalling the same reads twice. As shown in the runtime analysis earlier, SUP basecaller plus methylation Dorado model takes ~27 h, even with a NVIDIA A100 80GB GPU. DeepMod2 on the other hand, implements methylation detection that is separated from basecalling and re-uses basecalls with the help of move tables, taking only 6 h with a GPU and only 13 h with CPUs. As we continue to develop models for other kinds of methylations and modifications besides 5mC, DeepMod2 will also be able to assess various modifications in different motifs without resorting to re-basecalling for each of them. Additionally, the efficient implementation of DeepMod2 in terms of required computational resources and runtime also allows it to be used simultaneously with Guppy/Dorado methylation detection for consensus calling of methylation, as done by NANOME[27].

## Areas of further improvement in DeepMod2 and DNA methylation analyses
DeepMod2 uses signal summary statistics for each base as features for its deep-learning model, whereas Rockfish uses the actual signal value

itself; both tools rely on move tables to align signal to the basecalls. Move tables provide an approximate alignment between basecalls and signal, i.e. move table values denote when the basecaller prediction changed from one base to another in the flip-flop prediction model. Consequently, the move table does not always accurately demarcate boundaries between signals associated with consecutive bases, which can lead to uninformative signal summary statistics. Supplementary Fig. 9 shows signal-to-basecall alignment of three reads from R9.4.1 and R10.4.1 flowcell datasets of HG002 overlapping the same CpG locus. It can be seen in several cases that the signal associated to one base comprises two signal clusters towards the start and end of the signal, indicating poor signal alignment from move table. In such cases, the signal mean often just ends up in the center of the two clusters. The read signals also show that although the shape of the signal from different reads generally follows the same pattern, there can be substantial local timescale (x-axis) shifts and scaling differences in the signal alignments of different reads. These issues present a challenge for deep-learning models that need to be able to capture such variation in signals. Therefore, DeepMod2 uses several signal statistics, such as mean signal in each quadrant of base signal or median, to capture uncertainties in the basecall-to-signal alignment. Nevertheless, it is reasonable to expect that using the raw signal itself could potentially lead to more informative features and higher accuracy. This can be especially beneficial for rare modifications and methylations that may not be seen by the basecaller, and we will explore this direction in the future. However, for 5mC methylation detection, DeepMod2 is able to use signal summary statistics to accurately call methylation. Moreover, the pruned BiLSTM model of DeepMod2 has only 582,503 weights, compared to 4,367,110 weights of Rockfish "rf_small" model. A smaller model size means faster inference times, especially on CPUs.

Currently DeepMod2 only provides models for 5mCpG methylation because there is a lack of available datasets for other types of methylations. No 5hmC has been detected in GIAB cell lines HG001 through HG007, for which there is abundant Nanopore and bisulfite/oxidative bisulfite sequencing data available[34]. As more data for other types of modifications such as 5hmC and 6mA becomes available, we will continue to train DeepMod2 models for these modifications. It should be noted that although Dorado provides models for 6mA and 5hmC detection, as well as for 5mC detection in all genomic contexts, thus far no benchmarking study has evaluated or demonstrated the accuracy of these models.

In DeepMod2, we chose a probability threshold of 50% to label a per-read CpG prediction as methylated but it is possible to exclude per-read CpG predictions with intermediate probability scores to improve the accuracy of downstream analysis. This has been implemented by Nanopolish/f5c and Nanopore's Modkit tool for extracting per-site predictions from Guppy/Dorado BAM files. Although doing so can potentially improve the accuracy of per-site frequency estimates, the resulting decrease in coverage for per-site predictions can lead to lower statistical power in downstream analyses, especially for low coverage samples. Therefore, we do not recommend setting a stringent criterion to filter out per-read methylation predictions, and instead suggest that downstream tools should take confidence level of methylation calls into account within their analyses.

Finally, although differential methylation tools developed for bisulfite sequencing (BS-seq) can be used to analyze methylation calls from ONT, there is a need to comprehensively evaluate such tools on ONT methylation datasets and potentially develop new tools for downstream analysis of ONT methylation, such as NanoMethPhase[42]. Popular differential methylation analysis tools such as methylKit and DSS are designed for BS-seq and only use the counts of methylated and unmethylated cytosines in their statistical models. It can therefore be argued that using counts of binary methylation labels from BS-seq can lead to a lower statistical power compared to a robust method of integrating methylation probabilities from ONT methylation calling[42].

Moreover, BS-seq differential methylation detection tools analyze per-site methylation counts only and cannot incorporate the evidence of methylation from several CpG sites overlapping the same read over a span of thousands of bases, similar to the strategy used by Methyl-Purify for BS-seq[43]. Analyzing patterns of co-occurrence of variants over long reads has been successfully used to improve SNV detection in NanoCaller[35]. Therefore, it is reasonable to expect that incorporating read-level information from Nanopore sequencing for determining differential methylation can improve the accuracy and power of statistical analysis.

## Methods

### Oxford nanopore sequencing of NIH3T3 mouse genome

We sequenced NIH3T3 mouse cell line with Oxford nanopore sequencing using PromethION R9.4.1 (FLO-PRO002) flowcells and SQK-LSK110 library preparation kit. Sequencing was performed using standard Nanopore whole genome sequencing protocol and we generated 6.97 million reads and 79 Gbp bases at ~29X coverage, with mean, median and N50 read lengths of 11,324 bp, 6,694 bp and 21,744 bp respectively. NIH3T3 cell was purchased from ATCC.

### Reduced representation methylation sequencing of HG002

We performed reduced representation methylation sequencing (RRMS) on the human reference sample HG002. The genomic DNA sample was first sheared to about 6 kbp using the Covaris g-TUBE at a centrifuge speed of 11,00 RPM for 30 s. After shearing, 2 ug of the sheared DNA was processed into a Nanopore library using Nanopore's standard protocol for RRMS (ligation sequencing gDNA – reduced representation methylation sequencing of human samples: RRMS_9164_v110_revC_30-May2022). The library was processed using Nanopore's ligation sequencing kit (SQK-LSK110) and sequenced with a MinION Flow Cell (FLO-MIN106D) on the GridION. The sample was sequenced for 72 h with 2 washes and reloads at 23 h and 46 h. The BED file used for human RRMS can be accessed through ONT's "Adaptive Sampling Catalogue" (https://community.nanoporetech.com/adaptive_sampling_catalogue/). HG002 (NA24385) cell line was purchased from Coriell Institute for Medical Research.

### Datasets

**Benchmark datasets.** For HG002, HG003 and HG004, we obtained ground truth from EpiQC study[34], available from Gene Expression Omnibus (GEO) under the accession number GSE186383. For each genome, we downloaded CpG methylation calls from six short-read library preparation methods for cytosine deamination and whole genome sequencing, covering a range of deamination techniques. These methods include whole genome bisulfite sequencing (TruSeq, MethylSeq, SPLAT, TrueMethylBS), oxidative bisulfite sequencing (TrueMethylOX), and enzymatic deamination (EMSeq). For each library prep method, we summed methylated and unmethylated cytosine counts over its two replicates. For NIH3T3, we used the three untreated replicates from Sapozhnikov et al.[44], deposited at the GEO under the accession number GSE162138. The three replicates have following accession numbers: GSM4942823, GSM4942824 and GSM4942825. We also used Illumina Mouse Methylation BeadChip for benchmark evaluation on NIH3T3 from Lee et al.[45].

**Sequencing datasets.** For HG002, HG003 and HG004, we obtained R10.4.1 (4 kHz sampling) flowcell datasets from Oxford Nanopore Open Data Registry[46]. The R10.4.1 datasets of each genome were sequenced using two PromethION flowcells. R9.4.1 dataset of HG002 was obtained from Oxford Nanopore Open Data Project[46], whereas R9.4.1 dataset of HG003 and HG004 were obtained from Human Pangenome Reference Consortium[47]. The NIH3T3 genome was sequenced by us by ONT sequencing.

### DeepMod2 methylation detection framework

DeepMod2 takes ionic signals from Oxford Nanopore sequencing as input and carries out methylation prediction in two steps. (1) Per-read prediction: DeepMod2 extracts features and signal summary statistics from each read and uses a BiLSTM or Transformer model to predict methylation probability for each CpG site on a read. (2) Per-site prediction: DeepMod2 merges methylation predictions from all reads overlapping a CpG site and estimates the percentage of methylated reads. The details for both per-read and per-site predictions are described below.

### Per-read prediction

**Feature extraction.** To extract features from raw Nanopore signals of a read, DeepMod2 requires a mapping or alignment between basecalled sequence and raw signals. This signal alignment is inferred from "move table" generated by Guppy or Dorado basecaller. The details of feature extraction are described below and shown in Fig. 9:

1. Inputs: POD5/FAST5 signal files and a BAM file containing basecalled read sequences are the required input. The BAM file can be aligned or unaligned and it must contain move table if using POD5 file format or if the FAST5 files do not contain move table. Otherwise, move table from FAST5 file can be used if Guppy basecall group is specified. Aligned BAM file and reference genome FASTA file are optional but highly recommended to allow methylation calling on reference anchored CpG sites.

2. For each read:
   a. Obtain raw signal from FAST5/POD5 file and normalize it using median absolute deviation.
   b. Determine all loci on the read that match CG sequence motif, denote these sites as set A.
   c. Determine mapping orientation and loci of bases on the read that map to a CG motif in reference genome using BAM alignment file, and denote these read loci as set B. These loci on the read may not have CG motif due to basecalling error. This step is only performed if reference genome and aligned BAM file are provided.
   d. Take a union of read loci in sets A and B and generate features for these CpG loci.
   e. Users can set a read length or mean quality score threshold to filter out reads from being utilized. This can be useful for excluding short, rejected reads from RRMS adaptive sampling.

3. For each CpG locus on the read from the union of set A and B described above, create a 21×19 feature matrix for each base in a 21-base window centered at cytosine of interest:
   a. Use the move table from BAM file or FAST5 file to determine which segments of the normalized signal belong to each base in the 21-bp window.
   b. If the read is aligned and reference is provided, use CIGAR string from alignment to determine which reference base is mapped to each base in the read. The reference base aligned to a read base will be used as a feature in the deep learning model but can be excluded using '−exclude_ref_features' option. If '−exclude_ref_features' option is used or if a base on the read is unmapped, clipped or corresponds to an insertion, then we will use N as the reference base.
   c. Calculate the following 19 features for each signal segment associated with a base in the 21-bp window:

   1 – length of base signal (log10 base)
   2 – mean of base signal
   3 – standard deviation of base signal
   4 – mean of 1st quarter of base signal
   5 – mean of 2nd quarter of base signal
   6 – mean of 3rd quarter of base signal
   7 – mean of 4th quarter of base signal

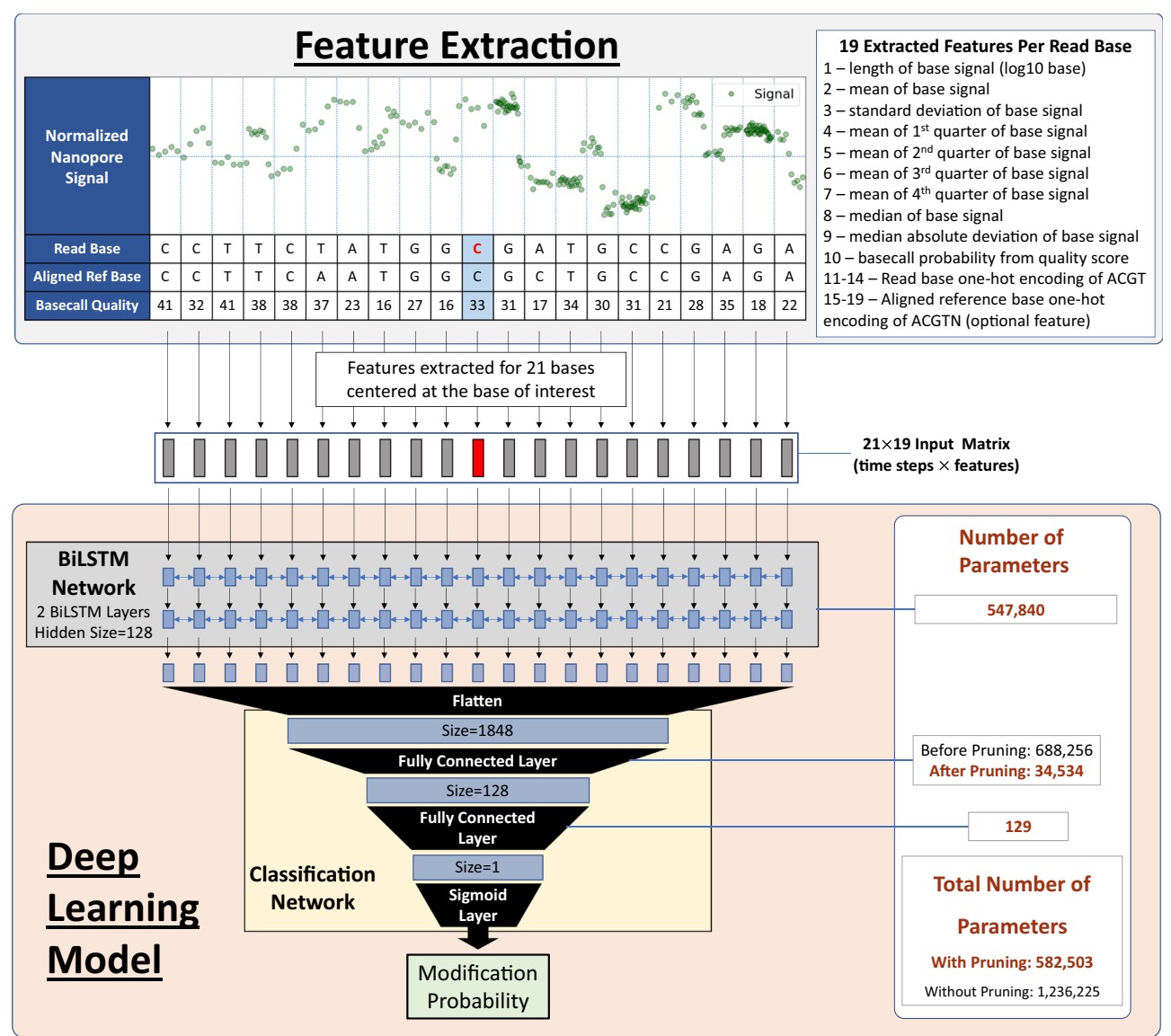

**Fig. 9 | DeepMod2 feature extraction and BiLSTM deep-learning model architecture.** For each CpG locus on a read, DeepMod2 extracts 19 features per read base in a 21-bp window centered at the cytosine of interest. The feature matrix is given as an input to a deep learning model, such as BiLSTM (shown in this figure) or Transformer, to predict methylation probability. DeepMod2 uses pruned neural networks by default to improve inference speed.

8 – median of base signal

9 – median absolute deviation of base signal

10 – basecall probability from quality score, converted from Phred scale to probability of correct basecall

11–14 – One-hot encoding of read base into ACGT

15–19 – One-hot encoding of aligned reference base into ACGTN (optional feature)

**Deep learning model inference.** After feature extraction, DeepMod2 feeds input matrix of each CpG site of each read into a BiLSTM or Transformer model to obtain a probability of 5mC methylation. DeepMod2 provides BiLSTM and Transformer model for various Nanopore flowcell chemistries. The details of the BiLSTM architecture are shown in Fig. 9, whereas the details of Transformer models are shown in Supplementary Fig. 10. Each BiLSTM model consists of two BiLSTM layers with 128 hidden units for 21 timesteps. The output of the BiLSTM model is flattened to $21 \times 128 \times 2 = 5376$ (timesteps × hidden size × bi-directional) size layer, which is fed into a fully connected layer with output size 128, followed by a fully

connected layer of output size 1 with sigmoid activation to obtain a probability score. This is in contrast with original DeepMod method which used BiLSTM output only from the central timestep for classification. By default, DeepMod2 uses pruned models for both BiLSTM and Transformer model, but this can be turned off using '--disable_pruning' parameter. Moreover, DeepMod2 will use GPU for model inference if available, otherwise it will resort to using CPUs. Users may also specify CPUs or a particular GPU device for inference using '--device' option. For each CpG feature matrix, DeepMod2 records read name, mapping locus and orientation of the CpG site on reference genome, methylation probability and prediction, read length and mean base quality score of the read, as well as the haplotype phase if the BAM file is phased. The above pieces of information for per-read predictions are stored in a tab separated text file. For CpG loci that match a cytosine in the read, we additionally add MM and ML tags to the BAM file to record the location and probability of modification. Since producing a compressed BAM output can be computationally expensive, users can specify number of threads to use for compression via '--bam_threads' parameter.

## Per-site prediction

As DeepMod2 produces per-read predictions, it combines them into per-site predictions simultaneously. A threshold of 0.5 for probability score is used to predict a CpG site as methylated, however, user specified thresholds can be used to exclude intermediate probability predictions from per-site methylation. DeepMod2 combines methylated and unmethylated predictions from all reads overlapping a CpG site in a strand-specific manner, i.e. it makes separate prediction for forward and reverse strand of reference genome. Then, it combines methylated and unmethylated counts from both strands to produce an aggregated per-site output as well. It provides additional breakdown of methylated and unmethylated counts for each haplotype if a phased BAM file was given as input. DeepMod2 determines per-site frequency as the fraction of reads containing methylated cytosines versus total number of mapped reads at the location (which can include SNVs but not deletions). By default, DeepMod2 only reports methylation frequencies for reference CpG sites, however '--include_non_cpg_ref' option can be used to get methylation frequencies for non-reference CpG sites as well (i.e. genomic loci that do not have CG motif in the reference but the sample has CG motif at that loci, potentially due to a variant). If the dataset of a sample is being split into multiple runs, then per-read predictions from all the runs can be combined later using the 'merge' function of DeepMod2. The merge function can also be used to re-calculate per-site frequencies with different modification thresholds or read quality score cutoff.

## Training and testing

**Training.** DeepMod2 models are implemented in PyTorch and are trained using Adam optimizer with a learning rate of 5e-5 and L2 regularization with a coefficient of 1e-5. In order to obtain ground truth label for model training, we used methylation predictions for HG002, HG003 and HG004 genomes from EpiQC study. The ground truth data consisted of methylation predictions from six short-read library methods for each genome (after adding up the two technical replicates for each method). For a CpG site to be included in the training dataset, it had to satisfy the following criteria in each of the six methods:

> Coverage ≥10
> Positive Label: ≥90% methylation
> Negative Label: <10% methylation

Therefore, a CpG site was labelled as positive if it had ≥90% methylation in all six methods and labelled as negative if it had <10% methylation in all six methods. We chose these strict criteria because we are training our model with supervised learning and only wanted to include those sites where we can confidently assume that all reads have the same methylation status. We used chr2-21 for training the model and chr22 for hyperparameter selection and validation, and we excluded chr1 from training or model validation. Since the methylation counts from EpiQC were aggregated from both forward and reverse strands of CpG sites, we split the CpG sites into forward and reverse strands. Afterwards, we were left 11.7 million positive and 7.3 million negative labels in chr2-21 training datasets of the three genomes combined; Supplementary Table 4 shows the number of positive and negative labels per genome in both training and validation datasets.

Once these labels were obtained, we use R9.4.1 and R10.4.1 flowcell datasets of HG002, HG003 and HG004 to generate the 21 × 19 feature matrices. For a given read and a CpG site, if the read sequence had less than 0.75 percent identity with aligned the reference sequence in the 21-bp window centered at the CpG site, we excluded that feature matrix from the final set of feature matrices. As a result, we generated a total of 635 million matrices for R10.4.1 datasets and 700 million matrices for R9.4.1 datasets. Since we wanted to train DeepMod2 models to be able to predict methylation calling from both aligned and unaligned reads, during the model training we used each feature matrix once with both read and reference sequence encodings included and once with only

read sequence encoded and reference sequence replaced by N, effectively doubling the number of training feature matrices to 1.3 billion and 1.4 billion for R10.4.1 and R9.4.1 models. We trained both BiLSTM and Transformer models for 10 epochs with early stopping. The BiLSTM models for R9.4.1 and R10.4.1 datasets were trained for 4 epochs (validation accuracy of 96.17%) and 5 epochs (validation accuracy of 97.89%), respectively, before the models started overfitting. Similarly, the Transformer models for R9.4.1 and R10.4.1 datasets were trained for 4 epochs (validation accuracy of 96.32%) and 5 epochs (validation accuracy of 97.95%), respectively.

**Testing.** For per-read evaluation, we created ground truth labels for chr1 of HG002, HG003 and HG004 from EpiQC dataset using the same criteria as the training labels since per-read evaluation is equivalent to testing model classification accuracy. For NIH3T3, we generated ground truth labels for autosomal chromosomes chr1-19 from three bisulfite sequencing replicates using the same criteria as human genomes, i.e. minimum coverage of 10 across the three replicates, with CpG sites labelled as positive or negative if they had ≥90% methylation or <10% methylation across the three replicates, respectively. Consequently, we had a total of 817305, 590693, 568969 and 151566 stranded CpG sites in per-site evaluation of HG002, HG003, HG004 and NIH3T3. A detailed breakdown is shown in Supplementary Table 5.

> For per-site evaluation, we loosened the criteria for positive and negative labels but still used the following criteria:
> For HG003-4: coverage ≥10 in all six short-read methods
> For NIH3T3: coverage ≥5 in all three BS-seq replicates
> Positive Label: ≥80% methylation in all six short-read methods for HG002-4 or three BS-seq replicates for NIH3T3.
> Negative Label: <20% methylation in all six short-read methods for HG002-4 or three BS-seq replicates for NIH3T3.

We chose 20% and 80% as thresholds for defining ground truth because EpiQC reported that sites with 20–80% methylation had poor concordance not only between ONT and short-reads method, but also among the different short-read methods. As a result, we decided to exclude these sites from evaluation since the ground truth labels for these sites cannot be reliably generated. We chose a smaller coverage threshold for NIH3T3 since the BS-seq data for it had lower coverage. Consequently, we had a total of 1438161, 1101209, 1042217, 9298784 stranded CpG sites in the per-site evaluation of HG002, HG003, HG004 and NIH3T3, shown in Supplementary Table 6.

For correlation analysis, we only used a coverage cutoff of 10 for HG002, HG003 and HG004 and cutoff of 5 for NIH3T3 to exclude low confidence CpG sites, and we did not use any filtering based on methylation frequencies. To get the final methylation frequency of reach CpG site, we averaged methylation levels over the six short-read methods from EpiQC for HG002-4 and averaged over the three BS-seq replicates for NIH3T3. At the end, we were left with ~4.5 million stranded CpG sites for HG002, HG003, HG004 and 15 million stranded CpG sites for NIH3T3. A detailed breakdown of number of CpG sites in per-site and correlation evaluation of each genome is shown in Supplementary Table 6.

To enable benchmarking in various genomic contexts, we annotated short-read ground truth methylation calls for the following features in GRCh38 and GRCm38: CpG islands, CpG shores, CpG shelves, exons, introns, promoters, and intergenic regions. To generate CpG islands, shores, and shelves annotations, we downloaded the GRCh38 CpG islands BED file and GRCm38 CpG islands BED file from the UCSC Genome Browser. To generate exons, introns, promoters, and intergenic regions, we downloaded the v43 gene annotation file (gencode.v43.annotation.gtf.gz) for GRCh38 and the vM25 gene annotation file (gencode.vM25.annotation.gtf.gz) for GRCm38 from GENCODE. CpG shores were generated by extending the region 2 kbp up- and downstream of the CpG islands and CpG shelves were generated from 2–4 kbp up- and downstream of the CpG islands. Exons were extracted

using the "exon" feature type from the GENCODE v43 annotation file for GRCh38 and GENCODE vM25 annotation file for GRCm38. Genes were extracted with the "gene" feature type. Introns were generated by using bedtools[48] subtract to take the difference between the genes and exons files. Promoters were generated by extending the region 1000 bp upstream of the transcription start sites. Intergenic regions were generated by using bedtools subtract to take the difference between the reference genomes and genes/promoters. The above annotations were then overlapped with all the CpG sites in each reference genome using bedtools intersect and were all compiled into an annotation bedgraph.

We compared the performance of DeepMod2 with Guppy, Dorado, Nanopolish, f5C and Rockfish. For Guppy we used "dna_r9.4.1_450bps_modbases_5mc_cg_sup" and "dna_r10.4.1_e8.2_400bps_modbases_5mc_cg_sup_prom" models implemented in Guppy V6.3.8 (accessed Nov 4 2022 from https://community.nanoporetech.com). For Nanopolish, we used v0.14.0 from Nanopolish's github repository (accessed Oct 12 2022), whereas for f5c, we used v1.3. We used Dorado v0.3.4 with the fast, hac and sup versions of "dna_r10.4.1_e8.2_400bps@v4.1.0" models, followed by Modkit for pileup. For Rockfish, we used github commit "079e7582c0fd8cb3017e37251f4d0105e94c0ecc", accessed on Aug 15 2023, and used "rf_small" model for inference.

### Haplotype specific differential methylation
We carried out SNV calling and phasing of R10.4.1 datasets of HG002, HG003 and HG004 with NanoCaller[35] v3.4.1, which uses WhatsHap[49] for phasing. Afterwards, we used DeepMod2 BiLSTM models to predict methylation in each parental haplotype of autosomal chromosomes chr1-22. For differential methylation detection, we used DSS[38] on each genome independently and provided methylation counts for each haplotype as two groups to be compared. We used the same DSS parameters for detection of differentially methylated regions as NanoMethPhase[42], except that we used stringent p-value cutoff of 1e-5 instead of 1e-3 for a differentially methylated CpG site in DSS to be considered statistically significant. For DMLtest function in DSS, we enabled smoothing, and set the delta threshold equal to 0 in callDMR function. Whereas for differentially methylated regions, we required them: (1) to be at least 100 bp long, (2) to contain at least 15 CpG sites, (3) at least 50% of CpGs in the region should be differentially methylated in statistically significant manner, and we allowed DSS to merge any differentially methylated regions within 100 bp of each other. We obtained a list of known ICRs by choosing the ICRs marked with "#" in Table 1 of Jima et al.[36] and obtained the list of all candidate ICRs from Supplementary Table S1 of Jima et al.[36], which can also be found at https://humanicr.org/.

### Reporting summary
Further information on research design is available in the Nature Portfolio Reporting Summary linked to this article.

## Data availability
ONT RRMS data of HG002 and ONT WGS data of NIH3T3 generated deposited in the BioProject under accession code PRJNA951714. EpiQC[34] ground truth for HG002, HG003 and HG004 is available from Gene Expression Omnibus (GEO) under the accession number GSE186383. For NIH3T3, bisulfite-sequencing ground truth[44] is available from GEO under the accession number GSE162138 and methylation microarray ground truth Lee et al[45]. is available in Supplementary Data 8 of this study. ONT R10.4.1 sequencing datasets of HG002, HG003 and HG004 are available from Oxford Nanopore Open Data Registry[46] under the following AWS storage bucket: s3://ont-open-data/giab_lsk114_2022.12/. ONT R9.4.1 dataset of HG002 is also available from ONT Open Data Registry [https://registry.opendata.aws/ont-open-data] under the following AWS storage bucket: s3://ont-open-data/gm24385_mod_2021.09/. HG003

and HG004 R9.4.1 ONT datasets from Human Pangenome Reference Consortium[47] are available from the following URLs: https://s3-us-west-2.amazonaws.com/human-pangenomics/index.html?prefix=NHGRI_UCSC_panel/HG003/nanopore/ and https://s3-us-west-2.amazonaws.com/human-pangenomics/index.html?prefix=NHGRI_UCSC_panel/HG004/nanopore/. BED file for human RRMS can be accessed through "Adaptive Sampling Catalogue" (https://community.nanoporetech.com/adaptive_sampling_catalogue/) from ONT's Nanopore Community page which requires customer login access. CpG islands BED files for GRCh38 and GRCm38 were downloaded from the UCSC Genome Browser using the following links, respectively: https://hgdownload.soe.ucsc.edu/goldenPath/hg38/database/cpgIslandExt.txt.gz and https://hgdownload.soe.ucsc.edu/goldenPath/mm10/database/cpgIslandExt.txt.gz. Gencode annotation files for GRCh38 (v43) and GRCm38 (vM25) were downloaded from the following links, respectively: https://ftp.ebi.ac.uk/pub/databases/gencode/Gencode_human/release_43/gencode.v43.annotation.gff3.gz and https://ftp.ebi.ac.uk/pub/databases/gencode/Gencode_mouse/release_M25/gencode.vM25.annotation.gff3.gz. Source data are provided with this paper.

## Code availability
The DeepMod2[50] software is available at https://github.com/WGLab/DeepMod2 and is distributed under the MIT License.

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

## Acknowledgements

This study is supported in part by NIH grant HG013359 (K.W.), CA279618 (K.W.), GM146978 (W.Z.), Penn Undergraduate Research Mentoring Program (K.W.), and the CHOP Research Institute (K.W.). We thank the IDDRC biostatistics and data science core supported by HD105354 for advice on machine learning. We would like to thank Dr. Qian Liu for support on DeepMod and for technical assistance with the development of DeepMod2 and thank Wonder Zhu for helping with performance evaluation.

## Author contributions

M.U.A. developed the computational method and implemented the software tool. M.U.A. and A.G. drafted the manuscript and evaluated performances. J.C. performed sequencing and analysis of RRMS. W.Z. advised on the study, provided guidance and materials. K.W. conceived the study, advised on model design, and guided implementation/evaluation. All authors read, revised, and approved the manuscript.

## Competing interests

The authors declare no competing interests.
