## [Peer Review File · Nature Communications]

A signal processing and deep learning framework for methylation detection using Oxford Nanopore sequencingREVIEWER COMMENTS

Reviewer #1 (Remarks to the Author):

Ahsan et al. developed DeepMod2, a deep-learning framework designed to detect DNA methylation, specifically 5-methylcytosine (5mC), using Oxford Nanopore Technologies sequencing, providing solutions to obstacles encountered with traditional techniques. The framework employs a sophisticated model to interpret ionic current signals during ONT sequencing, calculating the likelihood of 5mC methylation, and is adaptable to a range of data types and most latest flow cell technologies. As an open-source platform, DeepMod2 rivals some proprietary of methylation detection in both whole-genome and adaptive sequencing data.

However, in its current form, the reported results by this study don't stand up against tools such as Remora, and the proposed model seems less innovative than more advanced models such as Transformers and BERT. The proposed model also lacks the description of deep learning neural network structure, sharing numerous similarities with the previous version of DeepMod (10.1038/s41467-019-10168-2). Furthermore, I have the following concerns which should be addressed.

Primary comments:

1. The authors have built DeepMod2 on the foundations of the earlier DeepMod version (10.1038/s41467-019-10168-2), but there is scant information regarding the BiLSTM of DeepMod2. The net progress of DeepMod2 compared the previous version DeepMod, in terms of the algorithms and models, need to be clarified to provide the reader with a fulsome appreciation of the redesigned DeepMod2.

2. Recent works have also emphasized on deep learning neural network models for nanopore methylation calling software (e.g. 10.1109/BIBM52615.2021.9669841, 10.1101/2022.11.11.513492). These models have employed more advanced deep learning models (e.g., Transformers) compared to the BiLSTM used in this manuscript, which are capable of capturing a wider range of sequence features and signals. Could the authors provide their rationale for choosing BiLSTM over these newer models, and the reason why not compare with these advanced models?

3. The authors have stated that DeepMod2 supports 10.4 flowcells, yet the most advanced models in basecalling/methylation calling tools, such as Guppy and Dorado (<https://github.com/nanoporetech/dorado>), already support both 9.x and 10.x flowcells. It would be beneficial if the authors could underscore the unique aspects of their current version compared to

existing software. Additionally, reporting on the performance difference of DeepMod2 on 9.4 and 10.4 flowcells would make a significant contribution to ONT methylation calling studies.

4. It is surprising that one main advantage of nanopore sequencing over RRBS/WGBS, the ability to differentiate 5hmC from 5mC, has not been discussed or addressed in DeepMod2. Notably, tools such as Remora, Guppy, and Dorado possess this capability. Can the authors provide an explanation or discuss the potential challenges related to this omission?

Minor comments:

5. In Figure 3, the performance of DeepMod2 when used with Tombo and Guppy displays a starkly contrasting trend. The authors should give more emphasis to examining and comparing the results produced by both Tombo and Guppy. Specifically, in Fig. 3i, the Remora and Nanopolish model apply a default probability threshold that isn't 0.5, but rather 0.2/0.8, etc. Performance comparisons should also consider these alternative threshold suggestions with respect to DeepMod2.

6. In Figure 4, while it's commendable that the authors have demonstrated site-level performance in different genomic regions, the results seem to indicate that DeepMod2 doesn't perform as well as the Remora model across most of these regions. Strategies to improve performance in these challenging regions are of significant interest and need to be addressed.

7. In Figure 5, a comparison between not only R9.x and R10.x individually, but also both R9.x and R10.x together, would be informative. Moreover, the text in this figure is somewhat difficult to discern.

8. Figure 6 also demonstrates that the Remora and Nanopolish models show superior performance at the site level. The authors should explore strategies to further enhance DeepMod2's site-level performance, such as using a confidence threshold. Additionally, the total number of overlapping CpGs should be reported.

9. In Figure 7, a localized view of regions illustrates the comparison of ONT WGS and ONT RRMS. However, a broader, global view of these comparisons for each chromosome should also be presented to the readers.

10. In Figure 8, it's necessary to display the differential methylation regions of both cancer and normal samples, and also the regions at imprinting gene locations, for the sake of validation.

11. In Figure 9, the comparison of 5mC for RRMS and RRBS is insufficient. Including 5hmC would complement this figure.

12. In Table 2, a discussion should be provided on why Remora outperforms DeepMod2.

13. For Tables 3 and 4, it would be useful to include a report on DMRs (Differentially Methylated Regions) by chromosomes.

Reviewer #2 (Remarks to the Author):

Thank you for the opportunity to review the paper titled "A signal processing and deep learning framework for methylation detection using Oxford Nanopore whole-genome or adaptive sequencing" by Mian Umair Ahsan et al. DeepMod2 presents several advancements compared to DeepMod. These include the introduction of new models for ONT R10.4 flowcells, the ability to process both fast5 and pod5 files, the generation of per-read predictions, and the generation of a modified BAM file that enables visualization of methylation on IGV. In terms of performance, DeepMod2 demonstrates competitive results when compared to other methylation calling tools. Notably, the authors employed the new RRMS approach and evaluated the methylation prediction performance of DeepMod2 by comparing its methylation calls with ONT WGS using the HG002 sample and RRBS using paired breast cancer-normal samples.

Specific comments/questions:

1. In the Introduction (lines 111-113, it is mentioned that the HG002 cell line was sequenced using adaptive sampling to target 310Mbp, including CpG Islands, CpG shelves, CpG shores, and promoter regions. It would be helpful to specify the percentage of the genome targeted using adaptive sampling, as successful adaptive sampling typically targets 0.1% to 10% of the genome.

2. Regarding the DeepMod2 model, I have a couple of questions. Does the model filter out intermediate probability scores in per-read predictions? Additionally, in the post-processing step, it is mentioned that per-read predictions are filtered based on read quality score and length. Could you provide information

about the thresholds used for filtering? Can users customize these thresholds according to their requirements?

3. While DeepMod2 introduces significant updates compared to DeepMod, such as additional models, support for processing POD5 files, generating per-read predictions, and BAM alignment files with MM/MI tags, do DeepMod2 and DeepMod utilize the same BiLSTM model framework?

4. The paper mentions the performance benchmarking of four DeepMod2 models. Could you please elaborate on the differences between the DM2_Guppy_HG1_R9.4 and DM2_Guppy_HG2_R9.4 models?

5. You included fully methylated and fully unmethylated sites with significant coverage from BS-seq for per-read evaluation. What is the minimum coverage threshold you used for including these sites?

6. Remora is not a tool for methylation calling but provides an API to call modifications for ONT basecallers such Bonito, Guppy and Dorado. It would be better to specify the exact ONT basecaller used when employing the Remora model. I suggest replacing Remora with the specific methylation caller used (e.g., Guppy (remora)) consistently throughout the paper.

7. It is a bit hard to differentiate between each tool (the point shape) in Figure 3i and Figure 4, with the overlapping of the symbols making it hard to see. I recommend changing the scatter plot to a bar plot, utilizing the same color scheme as the ROC curve that shows the F1 score for each tool across different samples.

8. The paper mentions that all Nanopore methylation callers exhibited slightly worse performance for the mouse genome compared to human genomes. Since DeepMod2 performs similarly to ONT callers using the Remora model, what factors led the researchers to prefer DeepMod2 over ONT callers? To make DeepMod2 more unique, would you consider training it using the mouse genome to enable researchers to apply it to mouse species, given that most of the models were trained on human and E. coli?

9. RRMS employs adaptive sampling to target 310 Mb of the human genome, which is highly enriched for CpGs. To provide additional clarity, it would be beneficial to include information about the targeted regions used in adaptive sampling, such as the bed file and fasta file you provided to the MinKNOW. Perhaps the relevant files could be provided as Supplementary Data.

10. Have you compared the performance of DeepMod2 with other tools in terms of speed/running times and CPU/GPU usage using the same dataset? This information would assist researchers in selecting an appropriate methylation calling tool based on their specific needs and research purposes.

Minor comments:

1. In the Experimental Procedures (line 411), 2 wash should be corrected to “2 washes”
2. On line 211, per-read CpG are labelled as... should be capitalized as “Per-read CpG...”

RESPONSE TO REVIEWERS' COMMENTS

Summary

We thank the editors and reviewers for your constructive comments on our manuscript titled “A signal processing and deep learning framework for methylation detection using Oxford Nanopore sequencing”. We have taken these comments very seriously and we have made substantial efforts to address all concerns from the reviewers, while improving the manuscripts in several additional aspects not originally requested by the reviewers. With these additional changes during the course of the revision, we believe that the manuscript is substantially improved.

Several major changes in the revised manuscript include: (1) the display items in the previous version of the manuscript include data generated on both the R9.4.1 and R10.4.1 flowcells. Since Oxford Nanopore has officially deprecated R9 flowcells, and since we have access to more sequencing data (and the corresponding ground truth) on R10 flowcells, we focused the results on R10.4.1 flowcells and moved much of the R9.4.1 results to Supplementary Materials, but in several figures we still include a side-by-side comparison of these two technical platforms. To comprehensively present our findings on methylation calling, genomic imprinting, the inference of epihaplotypes, the handling of structural variants not in reference genome, we ended up with 12 Figures and 4 Tables in main text, in addition to many other supplementary tables and figures. (2) we have now added the transformer model to DeepMod2, in addition to the default BiLSTM model, as suggested by one reviewer. We have made extensive additional analysis and evaluations to compare the relative merits and to discuss model considerations. Similarly, perhaps due to our oversight, we did not explicitly compare DeepMod and DeepMod2 in terms of neural network structure and computational implementations; per one reviewer’s comments, we have now dedicated an entire subsection in Results to illustrate the differences and explain the design choices. (3) With the additional availability of several whole genome sequencing data, we re-trained the DeepMod2 models and observed large improvements: its performance is now similar to even the most recently released software by ONT (again, a closed-source tool). (4) We have now highlighted the methodological innovations even in the BiLSTM model so that readers have a better understanding of why and how it works, and under what circumstances that it may yield suboptimal results. The addition of transformer model into DeepMod2 is also innovative and is not described in the previous version of the manuscript.

We apologize for the long delay in submitting a revised manuscript. Despite this delay, we also feel that the extensive additional analysis, the inclusion of optional transformer models, as well as the restructuring of results from focusing on R9 to focusing on R10, are all well justified, not only to fully address the reviewers’ comments, but also to make the manuscript future-proof. Below are our point-by-point responses to the reviewers’ comments. The original reviewers’ comments are listed first, and our responses are colored in blue. The corresponding texts in the main texts are colored in red font to help the reviewers locate the changes in the paper that address their remarks directly. Thank you very much for your consideration of the revised manuscript.

Reviewer #1 (Remarks to the Author):

Ahsan et al. developed DeepMod2, a deep-learning framework designed to detect DNA methylation, specifically 5-methylcytosine (5mC), using Oxford Nanopore Technologies sequencing, providing solutions to obstacles encountered with traditional techniques. The framework employs a sophisticated model to interpret ionic current signals during ONT sequencing, calculating the likelihood of 5mC methylation, and is adaptable to a range of data types and most latest flow cell technologies. As an open-source platform, DeepMod2 rivals some proprietary of methylation detection in both whole-genome and adaptive sequencing data.

Author response: Thank you very much for the encouraging summary.

However, in its current form, the reported results by this study don't stand up against tools such as Remora, and the proposed model seems less innovative than more advanced models such as Transformers and BERT. The proposed model also lacks the description of deep learning neural network structure, sharing numerous similarities with the previous version of DeepMod (10.1038/s41467-019-10168-2). Furthermore, I have the following concerns which should be addressed.

Author response: Thank you very much for the insightful comments and feedback. We found them very helpful as we revised the methodology of DeepMod2. We added a detailed description of DeepMod2 BiLSTM model, which differs from DeepMod in several aspects that were not reflected in the previous version, and we included this comparison as a separate sub-section in the revised manuscript. As the reviewer has suggested, we have indeed been incorporating a transformer model in DeepMod2 (but did not mention it in previous version), and we have now performed a comparison between the two types of model architectures. Furthermore, since our previous submission, we retrained the models on a much larger dataset. Previously DeepMod2 models were trained on chr1 of HG002, but now we have retrained them on chr2-21 of HG002, HG003 and HG004 genomes. We also performed a deeper hyperparameter search to optimize DeepMod2 model architecture compared to the original DeepMod and added “model pruning” to further optimize the performance. We believe that with the updated models and results, DeepMod2 shows similar performance as Guppy (we previously termed it as Remora but changed it upon the second reviewer’s comments). We have also updated the evaluation to include more genomes and updated the benchmark to EpiQC so that the benchmark for all human genomes comes from the same source.

Primary comments:

1. The authors have built DeepMod2 on the foundations of the earlier DeepMod version (10.1038/s41467-019-10168-2), but there is scant information regarding the BiLSTM of DeepMod2. The net progress of DeepMod2 compared the previous version DeepMod, in terms of the algorithms and models, need to be clarified to provide the reader with a fulsome appreciation of the redesigned DeepMod2.

Author response: We appreciate these comments. DeepMod2 is not merely a rewritten version of DeepMod, but it includes many methodological differences that are not described in detail previously.

To fully address the reviewer's comments, we have added detailed description of DeepMod2 model in the main manuscript within "Methods" subsection "DeepMod2 Methylation Detection Framework" on pages 20-21, as well as in Figure 12 on page 37. "Supplementary File 2" Figure S6 also shows the details of DeepMod2 Transformer model. We have added a detailed comparison of DeepMod and DeepMod2 under "Results" subsection "Comparison of DeepMod2 with DeepMod" on page 12, as well as in "Supplementary File 2" Table S1. Performance comparison of DeepMod and DeepMod2 can be found in "Supplementary File 1" Table S9.

2. Recent works have also emphasized on deep learning neural network models for nanopore methylation calling software (e.g. 10.1109/BIBM52615.2021.9669841, 10.1101/2022.11.11.513492). These models have employed more advanced deep learning models (e.g., Transformers) compared to the BiLSTM used in this manuscript, which are capable of capturing a wider range of sequence features and signals. Could the authors provide their rationale for choosing BiLSTM over these newer models, and the reason why not compare with these advanced models?

Author response: We compared DeepMod2 performance against Rockfish as suggested here. (We only evaluated Rockfish on legacy R9.4.1 flowcell datasets as Rockfish cannot handle R10.4.1 flowcell.) We have indeed been incorporating a transformer model in DeepMod2, so that in the revised manuscript, we can present results from both transformers and BiLSTM models. Our results show that DeepMod2 Transformer model and Rockfish perform better than DeepMod2 BiLSTM model in terms of per-read performance. However, this does not necessarily translate into a higher per-site performance, where DeepMod2 BiLSTM model performs better than DeepMod2 Transformer model and performs similarly against Rockfish. Our analysis suggests that the performance of Rockfish may partially be due to that it uses raw signal as input for its model whereas DeepMod2 uses signal summary statistics. Indeed, Dorado and Guppy both use recurrent neural network for basecalling and methylation detection and show superb performance on R10.4.1 flowcells. Performance comparison between DeepMod2 and Rockfish can be found in "Per-read Evaluation" section on page 5, "Per-site Evaluation" section on pages 6-7, as well as in Figures 3- 6 on pages 28-31, and Tables 1-3 on pages 38-40. Supplementary File 1 Tables S1, S2 and S3 show detailed breakdown of performance comparison.

We chose not to include methBERT in our analysis since it requires Tombo which has been deprecated by Nanopore and only works for R9.4.1 datasets. Moreover, Tombo requires multi FAST5 to be split into single FAST5 files and then adds events to single FAST5 files, essentially tripling the storage requirement. This requires a substantial amount of storage and runtime, as the raw data for the genomes evaluated already consumed tens of terabytes of storage, even without running Tombo. As a result, we decided not to evaluate any tools that required Tombo as a pre-processing step, including DeepSignal and previously developed Tombo model in DeepMod2.

3. The authors have stated that DeepMod2 supports 10.4 flowcells, yet the most advanced models in basecalling/methylation calling tools, such as Guppy and Dorado (<https://github.com/nanoporetech/dorado>), already support both 9.x and 10.x flowcells. It would be beneficial if the authors could underscore the unique aspects of their current version compared to

existing software. Additionally, reporting on the performance difference of DeepMod2 on 9.4 and 10.4 flowcells would make a significant contribution to ONT methylation calling studies.

Author response: We added a detailed comparison of DeepMod2 with Guppy and Dorado in terms of accuracy and runtime performance under “Results” subsection “Evaluation of DeepMod2 Runtime and Accuracy Under Various Model Parameters, Basecaller and Alignment Options” on page 10, and under “Discussion” subsection “Notable features and advantages of DeepMod2” on pages 15-16. In short, we have argued that DeepMod2 provides an efficient and fast implementation of methylation calling that is separate from basecalling. Moreover, DeepMod2 framework provides quick runtime using both CPUs and GPUs. This can allow analysis of multiple modifications and motifs by resuing basecalls via move tables. On the other hand, Guppy and Dorado implement basecalling and methylation calling in a single step which requires high end GPUs, and running a different modification model requires re-doing the expensive basecalling step. Moreover, DeepMod2 now also allows a convenient method for analyzing haplotype-specific methylation calls without any extra runtime or steps.

4. It is surprising that one main advantage of nanopore sequencing over RRBS/WGBS, the ability to differentiate 5hmC from 5mC, has not been discussed or addressed in DeepMod2. Notably, tools such as Remora, Guppy, and Dorado possess this capability. Can the authors provide an explanation or discuss the potential challenges related to this omission?

Author response: Our study did not include 5hmC models due to lack of data available to train such models. We were unable to obtain training data from ONT even after multiple requests. Previous studies (<https://doi.org/10.1186/s13059-021-02529-2>), have shown that commonly used cell lines such as GIAB HG001-HG007 lack 5hmC which is the only data available to us. 5hmC is typically enriched in brain tissue, which are more difficult to obtain than standard commercially available cell lines that abundant only in 5mC. We would like to point out that although Dorado provides models for 5mC, 5hmC and 6mA detection in all genomic contexts, thus far no benchmarking study by ONT or elsewhere has evaluated or demonstrated the accuracy of these models, and no data has ever been made available on these modifications by ONT.

Minor comments:

5. In Figure 3, the performance of DeepMod2 when used with Tombo and Guppy displays a starkly contrasting trend. The authors should give more emphasis to examining and comparing the results produced by both Tombo and Guppy. Specifically, in Fig. 3i, the Remora and Nanopolish model apply a default probability threshold that isn't 0.5, but rather 0.2/0.8, etc. Performance comparisons should also consider these alternative threshold suggestions with respect to DeepMod2.

Author response: We decided to exclude Tombo as a pipeline step in DeepMod2 as it has now officially been deprecated by ONT. Moreover, the redesigned DeepMod2 models that use Guppy/Dorado basecalls perform much better than our previously trained Tombo model, so we decided not to complicate the study design by focusing on performance obtained using a deprecated tool. Figure 3 pertains per-read performance, where ROC (Receiver Operating Characteristic) curve and Precision-Recall (PR) curve show the model predictions change as a “single decision threshold” changes. Using two

separate thresholds for positive and negative labels is therefore not compatible with ROC and PR analysis. Per-read performance measures how well the underlying probability model of a tool is, for which it is necessary to include all probability outputs. With regards to Remora/Guppy and Nanopolish, the exclusion of intermediate probabilities only comes into play when combining per-read predictions into per-site predictions. As a result, when we analyze per-site results of these tools in Figure 4 on page 29, we used the suggested or default settings of these tools for excluding intermediate probabilities. Whereas for DeepMod2, we do not recommend excluding intermediate probabilities. The main reason is that removing intermediate probability prediction might improve per-site frequency “percentage of methylation” estimates, but it also decreases the overall coverage. This can worsen the quality and statistical power of downstream analyses for low coverage samples.

6. In Figure 4, while it's commendable that the authors have demonstrated site-level performance in different genomic regions, the results seem to indicate that DeepMod2 doesn't perform as well as the Remora model across most of these regions. Strategies to improve performance in these challenging regions are of significant interest and need to be addressed.

Author response: Thank you for bringing up this comment. As we have explained earlier, using updated DeepMod2 models, we are able to show that DeepMod2 performs substantially better than Guppy on R9.4.1 flowcells, and it performs on par with the latest version of Guppy and Dorado on R10.4.1 flowcells. The updated Figure 4 is shown on page 29.

7. In Figure 5, a comparison between not only R9.x and R10.x individually, but also both R9.x and R10.x together, would be informative. Moreover, the text in this figure is somewhat difficult to discern.

Author response: We have simplified the evaluation so that now we have same three genomes (HG002, HG003 and HG004) available in both R9.4.1 and R10.4.1 datasets. We decided to remove HG001 dataset from evaluation since we did not have R10.4.1 dataset available for it. The comparison of R9 vs R10 evaluated against ground truth dataset of three genomes illustrates the improvements in R10 flowcells over R9. We did not add heatmaps of R9 against R10 since the figure already contains too many subfigures; additionally, we also increased the font size of the text. The updated figure is shown on page 30.

8. Figure 6 also demonstrates that the Remora and Nanopolish models show superior performance at the site level. The authors should explore strategies to further enhance DeepMod2's site-level performance, such as using a confidence threshold. Additionally, the total number of overlapping CpGs should be reported.

Author response: The new models in DeepMod2 show vastly improved performance over Nanopolish and perform similarly against Remora/Guppy. We adjusted the plots to include the number of overlapping CpGs in Figure 6 on page 31.

9. In Figure 7, a localized view of regions illustrates the comparison of ONT WGS and ONT RRMS. However, a broader, global view of these comparisons for each chromosome should also be presented to the readers.

Author response: Thank you for the suggestion. In the revised manuscript, Figure 7 i) on page 32 and “Supplementary File 3” show views at both a global and chromosome-level scale of ONT RRMS vs ONT WGS coverage and methylation frequencies.

10. In Figure 8, it's necessary to display the differential methylation regions of both cancer and normal samples, and also the regions at imprinting gene locations, for the sake of validation.

Author response: We decided to remove the analysis of RRMS vs RRBS for differential methylation detection in the five replicates of normal-cancer cell line samples. We believe that a rigorous evaluation of how the differentially methylated regions between the two technologies differ is out of scope of the current study, which highlights how DeepMod2 can enhance various aspects of DNA methylation detection (differential methylation requires third-party tool and a separate statistical model). However, we did include a section on how haplotype-specific methylation calling from DeepMod2, combined with our previously developed NanoCaller tool, can allow detection of imprinted regions. This analysis can be found in the main manuscript within “Results” subsection “Haplotype Specific Methylation Calling” on page 9 and Table 4 on page 41 where we compare DeepMod2 results against well know imprinted regions. We also provided a complete list of candidate imprinted regions found using DeepMod2 methylation calls from HG002, HG003, HG004 in “Supplementary File 4”.

11. In Figure 9, the comparison of 5mC for RRMS and RRBS is insufficient. Including 5hmC would complement this figure.

Author response: We decided to exclude RRMS vs RRBS comparison as pointed out in the previous comment. Moreover, the oxidative bisulfite sequencing data for the HHC1395/HCCBL1395 cell lines is not available, so our analysis cannot be backed up by a ground truth.

12. In Table 2, a discussion should be provided on why Remora outperforms DeepMod2.

Author response: We have included detailed discussion of differences between Guppy and DeepMod2 towards the end of “Per-site Evaluation” subsection of “Results” on page 7, and in “Supplementary File 2” Figure S1. Due to the use of updated models, DeepMod2 performs similarly against Guppy now.

13. For Tables 3 and 4, it would be useful to include a report on DMRs (Differentially Methylated Regions) by chromosomes.

Author response: We replaced this analysis with haplotype-specific methylation analysis and imprinted region detection, to show how paternal chromosomes differ from maternal chromosomes through haplotype-specific methylation analysis. This analysis can be found in the main manuscript within "Results" subsection "Haplotype Specific Methylation Calling" on page 9 and Table 4 on page 41 where we compare DeepMod2 results against well know imprinted regions. We also provided a complete list of candidate imprinted regions found using DeepMod2 methylation calls from HG002, HG003, HG004 in "Supplementary File 4".

Reviewer #2 (Remarks to the Author):

Thank you for the opportunity to review the paper titled "A signal processing and deep learning framework for methylation detection using Oxford Nanopore whole-genome or adaptive sequencing" by Mian Umair Ahsan et al. DeepMod2 presents several advancements compared to DeepMod. These include the introduction of new models for ONT R10.4 flowcells, the ability to process both fast5 and pod5 files, the generation of per-read predictions, and the generation of a modified BAM file that enables visualization of methylation on IGV. In terms of performance, DeepMod2 demonstrates competitive results when compared to other methylation calling tools. Notably, the authors employed the new RRMS approach and evaluated the methylation prediction performance of DeepMod2 by comparing its methylation calls with ONT WGS using the HG002 sample and RRBS using paired breast cancer-normal samples.

Author response: Thank you very much for the summary.

Specific comments/questions:

1. In the Introduction (lines 111-113, it is mentioned that the HG002 cell line was sequenced using adaptive sampling to target 310Mbp, including CpG Islands, CpG shelves, CpG shores, and promoter regions. It would be helpful to specify the percentage of the genome targeted using adaptive sampling, as successful adaptive sampling typically targets 0.1% to 10% of the genome.

Author response: Thank you for bringing up this issue. We have specified the percentage targeted (10%) in the Introduction on page 4.

2. Regarding the DeepMod2 model, I have a couple of questions. Does the model filter out intermediate

probability scores in per-read predictions? Additionally, in the post-processing step, it is mentioned that per-read predictions are filtered based on read quality score and length. Could you provide information about the thresholds used for filtering? Can users customize these thresholds according to their requirements?

Author response: In DeepMod2 results shown in the main manuscript, we did not apply any filter on intermediate probability score, read lengths and quality score, however all of these filters have thresholds that can be set by the user. By default, DeepMod2 does not apply threshold to exclude intermediate probability score, i.e. the probability threshold for modification is $<50\%$ and $\geq 50\%$. Similarly, no threshold is set of read length or quality, but a minimum read length of 600bp is recommended for RRMS to exclude off-target reads from analysis.

3. While DeepMod2 introduces significant updates compared to DeepMod, such as additional models, support for processing POD5 files, generating per-read predictions, and BAM alignment files with MM/MI tags, do DeepMod2 and DeepMod utilize the same BiLSTM model framework?

Author response: Thank you for asking the questions. DeepMod and DeepMod2 have different BiLSTM model architectures. We have added detailed description of DeepMod2 BiLSTM model in the main manuscript within “Methods” subsection “DeepMod2 Methylation Detection Framework” on pages 20-21, as well as in Figure 12 on page 37. “Supplementary File 2” Figure S6 also shows the details of DeepMod2 Transformer model. We have added a detailed comparison of DeepMod and DeepMod2 under “Results” subsection “Comparison of DeepMod2 with DeepMod” on page 12, as well as in “Supplementary File 2” Table S1. Performance comparison of DeepMod and DeepMod2 can be found in “Supplementary File 1” Table S9. In short, original DeepMod analyzes Nanopore signal using three BiLSTM layers of size 100 followed by a single fully connected layer that processed only the middle timestep of the BiLSTM output. DeepMod2 on the other hand analyzes Nanopore signal using two BiLSTM layers of size 128, and applies a fully connected layer to the entire BiLSTM output for methylation probability calculation. Moreover, DeepMod2 applies model pruning to allow efficient inference and also provides Transformer models. The original DeepMod also applied an additional BiLSTM to update methylation frequencies based on methylation levels of nearby sites, whereas DeepMod2 does not use any such network.

4. The paper mentions the performance benchmarking of four DeepMod2 models. Could you please elaborate on the differences between the DM2_Guppy_HG1_R9.4 and DM2_Guppy_HG2_R9.4 models?

Author response: We appreciate the comments and we understand the confusion this has caused. We have updated and simplified DeepMod2 models in the revised version of the manuscript. Now we present only one R9 And R10 model for BiLSTM and Transformer models. Each model is trained on chr2-21 of HG002, HG003 and HG004 combined, and we decided not to train separate models on different genomes. The details of model training can be found in Methods subsection “Training and Testing” subsection on page 22.

5. You included fully methylated and fully unmethylated sites with significant coverage from BS-seq for per-read evaluation. What is the minimum coverage threshold you used for including these sites?

Author response: We used minimum coverage of 10 of per-read and per-site evaluation, and correlation analysis of HG002, HG003 and HG004. For NIH3T3, we used minimum coverage of 10 for per-read evaluation, but for per-site evaluation and correlation analysis we decreased the cutoff to 5. We have include these details in the Results section on pages 6 and 7, and Methods subsection “Training and Testing” on pages 22-23.

6. Remora is not a tool for methylation calling but provides an API to call modifications for ONT basecallers such Bonito, Guppy and Dorado. It would be better to specify the exact ONT basecaller used when employing the Remora model. I suggest replacing Remora with the specific methylation caller used (e.g., Guppy (remora)) consistently throughout the paper.

Author response: We have replaced the term Remora with Guppy and Dorado throughout the paper to be more consistent with the current convention.

7. It is a bit hard to differentiate between each tool (the point shape) in Figure 3i and Figure 4, with the overlapping of the symbols making it hard to see. I recommend changing the scatter plot to a bar plot, utilizing the same color scheme as the ROC curve that shows the F1 score for each tool across different samples.

Author response: Thank you for the suggestion. We have updated Figure 3 and 4 on pages 28 and 29 to use bar plots instead.

8. The paper mentions that all Nanopore methylation callers exhibited slightly worse performance for the mouse genome compared to human genomes. Since DeepMod2 performs similarly to ONT callers using the Remora model, what factors led the researchers to prefer DeepMod2 over ONT callers? To make DeepMod2 more unique, would you consider training it using the mouse genome to enable researchers to apply it to mouse species, given that most of the models were trained on human and E. coli?

Author response: We updated the evaluation strategy to exclude ground truth CpG sites with methylation frequency between 20-80%. In the revised manuscript, the ground truth uses more reliable methylated and unmethylated sites for evaluation, whereas previously we used 50% threshold to define methylated and unmethylated sites in the ground truth. We changed this threshold based on the suggestions of the EpiQC study (<https://doi.org/10.1186/s13059-021-02529-2>). After this change, we see that NIH3T3 performance more closely resembles performance on human genomes. In the future,

we will consider training models on mouse genomes, or train models on multiple organisms all together, once R10.4.1 flowcell dataset of NIH3T3 is available, either generated by us or by others.

9. RRMS employs adaptive sampling to target 310 Mb of the human genome, which is highly enriched for CpGs. To provide additional clarity, it would be beneficial to include information about the targeted regions used in adaptive sampling, such as the bed file and fasta file you provided to the MinKNOW. Perhaps the relevant files could be provided as Supplementary Data.

Author response: In the revised manuscript, we clarified that we used human RRMS BED file provided by ONT (https://community.nanoporetech.com/adaptive_sampling_catalogue/).

10. Have you compared the performance of DeepMod2 with other tools in terms of speed/running times and CPU/GPU usage using the same dataset? This information would assist researchers in selecting an appropriate methylation calling tool based on their specific needs and research purposes.

Author response: We have added runtime and resource comparison under “Results” subsection “Evaluation of DeepMod2 Runtime and Accuracy Under Various Model Parameters, Basecaller and Alignment Options” on pages 10-11.

Minor comments:

1. In the Experimental Procedures (line 411), 2 wash should be corrected to “2 washes”
2. On line 211, per-read CpG are labelled as... should be capitalized as “Per-read CpG...”

Author response: Thank you for the comments. We have updated these terms.

REVIEWERS' COMMENTS

Reviewer #1 (Remarks to the Author):

Ahsan et al. have presented an updated version of their deep learning framework, DeepMod2, aimed at enhancing methylation detection using Oxford Nanopore sequencing. The addition of the Transformer model is a significant advancement, and the support for both POD5 and FAST5 formats broadens the applicability of this tool. It's commendable that the team evaluated the framework on both R9.4 and R10 flowcells, ensuring a comprehensive assessment of its performance. Comparing DeepMod2's performance with other state-of-the-art deep learning models is an essential step, and I appreciate the thoroughness in this analysis. Moreover, the detailed elaboration on the progress made from DeepMod to DeepMod2 provides valuable insights into the evolution of this tool. The current version of revised manuscript meets my expectations satisfactorily.

Reviewer #2 (Remarks to the Author):

The authors have fully addressed all of my major comments and I now support publishing this manuscript.

RESPONSE TO COMMENTS

Reviewer #1 (Remarks to the Author):

Ahsan et al. have presented an updated version of their deep learning framework, DeepMod2, aimed at enhancing methylation detection using Oxford Nanopore sequencing. The addition of the Transformer model is a significant advancement, and the support for both POD5 and FAST5 formats broadens the applicability of this tool. It's commendable that the team evaluated the framework on both R9.4 and R10 flowcells, ensuring a comprehensive assessment of its performance. Comparing DeepMod2's performance with other state-of-the-art deep learning models is an essential step, and I appreciate the thoroughness in this analysis. Moreover, the detailed elaboration on the progress made from DeepMod to DeepMod2 provides valuable insights into the evolution of this tool. The current version of revised manuscript meets my expectations satisfactorily.

Author response: Thank you very much for the summary.

Reviewer #2 (Remarks to the Author):

The authors have fully addressed all of my major comments and I now support publishing this manuscript.

Author response: Thank you very much for the summary.